# Anomalous 2022 deep water formation and intense phytoplankton bloom in the Cretan area

Anna Teruzzi[1], Ali Aydogdu[2], Carolina Amadio[1], Emanuela Clementi[2], Simone Colella[3], Valeria Di Biagio[1], Massimiliano Drudi[4], Claudia Fanelli[3], Laura Feudale[1], Alessandro Grandi[4], Pietro Miraglio[2], Andrea Pisano[3], Jenny Pistoia[2], Marco Reale[1], Stefano Salon[1], Gianluca Volpe[3], Gianpiero Cossarini[1]

[1]National Institute of Oceanography and Applied Geophysics - OGS, Trieste, 34142, Italy
[2]Ocean Modelling and Data Assimilation Division, Euro-Mediterranean Centre on Climate Change (CMCC) Foundation, 40127, Bologna, Italy
[3]Istituto di Scienze Marine – Consiglio Nazionale delle Ricerche, Roma, 00133, Italy
[4]Ocean Predictions and Applications Division, Euro-Mediterranean Centre on Climate Change (CMCC) Foundation, 73100 Lecce, Italy

*Correspondence to*: Anna Teruzzi (ateruzzi@ogs.it)

**Abstract.** The Mediterranean Sea is a quasi-permanently stratified and oligotrophic basin with intense late-winter early-spring phytoplankton blooms typically limited to few regions (i.e., northwestern Mediterranean Sea, the southern Adriatic Sea and the Rhodes gyre). In these areas, blooms are sustained by nutrients injection to surface layers by winter vertical mixing and convective processes. A markedly intense bloom was predicted in spring 2022 in an unusual area of the southeastern Mediterranean Sea (i.e., southeastern of Crete) by the MED-MFC system, the production centre of the Copernicus Marine Service for the Mediterranean Sea. Combining Copernicus modelling and observation products, the 2022 event and a number of driving and concurrent features have been investigated in a multidisciplinary way. A noticeable cold spell that occurred in eastern Europe at the beginning of 2022 has been identified as the main driver of an intense deep water formation event, with associated high nutrient concentrations in the surface layers. Consequently, an extreme phytoplankton bloom that was 50% more intense than usual occurred in the area southeast of Crete, starting nearly one month later than usual and lasting for 3-4 weeks. Impacts on primary production were also relevant in the 2022 event area, being 35% higher than the climatological annual primary production. Further, the documented link between primary productivity and fish catches suggests possible consequences along the whole food chain up to the marine ecosystem in the eastern Mediterranean Sea.

## 1 Introduction

The Mediterranean, the main regional sea of southern Europe, is a semi-enclosed basin located in a transitional zone between the midlatitude and subtropical climate regimes (Coppini et al., 2023; Cossarini et al., 2019; Lazzari et al., 2012; Siokou-Frangou et al., 2010). The Mediterranean Sea is an almost permanently stratified and oligotrophic basin with a few areas exhibiting recurrent late-winter/early-spring phytoplankton blooms: the northwestern Mediterranean, the southern Adriatic

Sea, and the Rhodes gyre (Siokou-Frangou et al., 2010). In these regions, the blooms are driven by deep winter convective processes, which brings nutrients into the surface layer, and by the subsequent stratification, when the phytoplankton is no longer diluted across the water column. At this moment, conditions are suitable for the surface phytoplankton bloom onset since in the surface layer both light and nutrients are available (Habib et al., 2022; Mayot et al., 2017; D'Ortenzio and Ribera d'Alcalà, 2009).

The Eastern Mediterranean experiences particularly sharp oligotrophic conditions, and productive areas are limited to the Rhodes gyre, where deep water mixing and related bloom events typically occur (Varkitzi et al., 2020), and to the Turkish coast (Kubin et al., 2019).

Current evidence suggests that the Mediterranean Sea is facing an increase in marine heat waves and a decrease in cold spell events (Simon et al., 2022) as a potential consequence of changes in regional climate. However, in March 2022 a strong and unusual atmospheric cold spell affected the eastern Mediterranean region (Demirtaş, 2023), with strong surface air temperature negative anomalies recorded over southeastern Europe (up to -3 °C according to Copernicus Climate Change Service bulletin; C3S monthly climate bulletin explorer, 2023; Surface air temperature for March 2022 | Copernicus, 2023).

In the present work, implications on marine physical and biogeochemical dynamics of the unusual 2022 cold event are investigated exploiting the products of the Copernicus Marine Service (Home | CopernicusMarine, 2023). We use both models and observations to highlight the interplay between biogeochemical and physical processes considering that intense cold spells usually drive deep water column mixings and consequent nutrients injections in the surface layer and onset of phytoplankton blooms (Auger et al., 2014). In order to describe the exceptionality of the 2022 event and its possible implications on the Mediterranean ecosystem, its spatial and temporal extent are defined based on phytoplankton bloom anomaly, and its characteristics in terms of sea surface temperature, mixed layer depth, surface chlorophyll, nutrient concentrations and primary production are investigated.

## 2 Methods

The occurrence and mechanism driving the anomalous deep convection and phytoplankton bloom episode southeast of Crete (eastern Mediterranean Sea) in spring 2022 was investigated using both model and satellite-based products. The Copernicus Marine Service Mediterranean monitoring and forecasting centre (MED-MFC) provides 3D ocean biogeochemical and physical variables at 1/24 ° resolution (product ref. 1, 2, 4 and 5; Table 1). Sea surface temperature (SST) and surface layer chlorophyll concentration are provided by the Copernicus Marine Sea Surface Temperature and Ocean Colour Thematic Assembly Centre (SST TAC and OC TAC, respectively) (Product ref. 3 and 6; Table 1). Both near real time and multi-year products are used to characterise the 2022 event.

| Product ref. no. | Product ID & type | Data access | Documentation |
|---|---|---|---|
| 1 | MEDSEA_ANALYSISFORECAST_BGC_006_014, Numerical models | EU Copernicus Marine Service Product (2022a) | Quality Information Document (QUID): Feudale et al. (2022), Product User Manual (PUM): Lecci et al. (2022a) |
| 2 | MEDSEA_MULTIYEAR_BGC_006_008, Numerical models | EU Copernicus Marine Service Product (2022b) | Quality Information Document (QUID): Teruzzi et al. (2022), Product User Manual (PUM): Lecci et al. (2022b) |
| 3 | OCEANCOLOUR_MED_BGC_L3_MY_009_143, Satellite observations | EU Copernicus Marine Service Product (2022c) | Quality Information Document (QUID): Colella et al. (2022a), Product User Manual (PUM): Colella et al. (2022b) |
| 4 | MEDSEA_ANALYSISFORECAST_PHY_006_013, Numerical models | EU Copernicus Marine Service Product (2022d) | Quality Information Document (QUID): Goglio et al. (2022), Product User Manual (PUM): Lecci et al. (2022c) |
| 5 | MEDSEA_MULTIYEAR_PHY_006_004, Numerical models | EU Copernicus Marine Service Product (2022e) | Quality Information Document (QUID): Escudier et al. (2022), Product User Manual (PUM): Lecci et al. (2022d) |
| 6 | SST_MED_SST_L4_REP_OBSERVATIONS_010_021, Satellite observations | EU Copernicus Marine Service Product (2022f) | Quality Information Document (QUID): Pisano et al. (2022a), Product User Manual (PUM): Pisano et al. (2022b) |
| 7 | OMI_VAR_EXTREME_WMF_MEDSEA_area_averaged_mean, Numerical models | EU Copernicus Marine Service Product (2023g) | Quality Information Document (QUID): Lyubartsev et al. (2023), Product User Manual (PUM): Lyubartsev and Clementi (2022) |
| 8 | ECMWF AST | ECMWF: IFS Documentation CY47R3, E.: IFS Documentation CY47R3 https://doi.org/10.21957/eyrpir4vj, 2021. | ECMWF (2021) |

**Table 1. Datasets used in the present work, with references and doi.**

A daily climatology based on the 1999-2019 MED-MFC Biogeochemistry Reanalysis (Cossarini et al., 2021; Teruzzi et al., 2022; product ref. 2, Table 1) was calculated following Hobday et al. (2016) for a subset of variables namely surface chlorophyll concentration, phosphocline depth, average phosphate concentration above the phosphocline, primary production integrated in the 0-200 m layer. The nutrient analysis was focused on phosphate since it is known as the limiting nutrient for the Mediterranean Sea (Siokou-Frangou et al., 2010). For each variable a set of percentiles is calculated to identify specific thresholds (i.e., 1[st], 25[th], 50[th], 75[th], 99[th] percentile) using a ten-day window centred on each date of the climatological year. Comparing the 2022 MED-MFC Analysis and Forecast (Salon et al., 2019; Feudale et al., 2022; product ref. 1, Table 1) with the corresponding climatology in the time window of the bloom (20 March - 30 April), the chlorophyll concentrations in all the surface grid points of the investigated area (22-32 °E, 32-35 °N) resulted above the 99[th] percentile for at least 20% of the time-window, indicating that the whole area was interested by intense and anomalous bloom conditions. In order to define the

region mostly interested by the anomalous 2022 bloom, following Hobday et al. (2016), the maximum difference with respect to climatology between 20 March and 30 April is calculated (Imax) and the event area is defined as composed by all the surface grid points with Imax higher than its 90th percentile (0.23 mg chl m$^{-3}$) (Fig. 1). The characteristics of the anomalous convection

and bloom event are investigated considering marine physical and biogeochemical properties averaged over the event area contoured in Fig. 1. Moreover, a daily sea surface temperature (SST) climatology derived from the Mediterranean SST multi-year satellite product over the period 1982-2021 (Pisano et al., 2022a, 2022b; product ref. 6, Table 1) is used to compare modelled surface temperatures (i.e., first layer of the MED-MFC SST; product ref. 4) during the 2022 event time window. To compare the physical and biogeochemical dynamics in the event area and in the Rhodes gyre area, vertical profiles of

temperature, phosphate and chlorophyll are investigated at two locations (Fig. 1).

Finally, the Copernicus Marine Ocean Monitoring Indicator (Lyubartsev et al., 2023; product ref.7, Table 1), computed from the Mediterranean Sea Physics Reanalysis (Escudier et al., 2021; product ref. 6, Table 1), which provides water mass formation rates in the Mediterranean Sea is considered to analyse the exceptionality of the 2022 event. In particular the Levantine Deep Water (LDW) formation index is calculated for 2022 and compared with values that occurred in the past (from 1987 onwards).

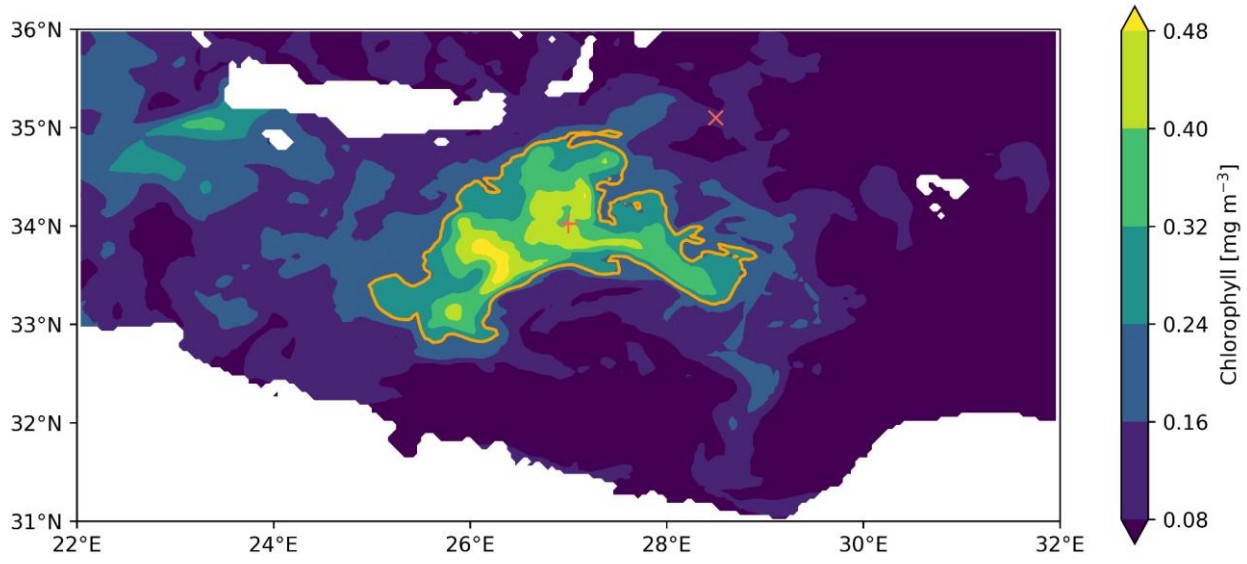

**Figure 1: Mediterranean Sea Analysis and Forecast product (product ref. 1, Table 1): filled contours of the maximum intensity of surface chlorophyll concentration between 20 March and 30 April 2022, and line contour of the 90th percentile of the maximum intensity (0.23 mg chl m$^{-3}$). The "+" and "x" markers indicate positions inside and outside the 2022 event.**

### 3 Results

The signal of the 2022 cold outbreak over Eastern Europe (Demirtaş, 2023) is clearly detectable in the atmospheric surface temperature (AST) extracted from the ECMWF analysis products (ECMWF 2021; product ref. 8, Table 1). In the second half of January 2022 the AST daily time series reaches a relative minimum (nearly 15° C), and it is followed by two minima in the

March-April time window (Fig. 2a). Accordingly, with a less noticeable variability, sea surface temperature (SST) gradually decreases in the area of interest (Fig. 2a). The satellite SST (Pisano et al., 2022a; product ref. 6, table 1) shows a constantly negative anomaly of winter 2022 with respect to its climatology (-0.46 °C on average) from the beginning of January to the end of March, indicating the sea surface cooling as the most likely driver of an anomalous deep convection event. Even lower SST values are provided by the MED-MFC Analysis and Forecast (Goglio et al., 2022; product ref. 4, Table 1) with a minimum in the second half of March, followed by a relatively sharp increase towards the SST satellite climatology. According with the relatively low SST and similarly to the typical winter mixing conditions in the Rhodes gyre area (Kubin et al., 2019), in the 2022 event area (Fig. 1) the mean mixed layer (MLD; calculated as depth where the density increases by 0.01 kg m$^{-3}$ compared to density at 10 m depth; product ref. 4, Table 1) is deeper than 500 m (Fig. 2b) on several occasions from the end of January to the end of March, when the mean MLD gets shallower (up to 50 m). Consistently with the strong March 2022 sea surface cooling, the mean MLD reaches its maximum in March (equal or deeper than 700 m). The daily maps of AST, SST, SST anomaly, MLD and heat fluxes during March 2022 provided in the Appendix A (Figs. A1-A5) further detail the spatial extent and temporal sequence of the atmospheric and oceanic processes summarised in Fig. 2. Two close significant drops in AST are, in fact, observed in the area (11-14 March and 19-23 March) according to a cold air intrusion from the northwest (Fig. A1). Together with the January cold spell (Fig. 2), the March cooling events resulted in significant negative SST anomalies especially south of Crete, which persisted in the area till the end of March (Fig. A2) with more steady occurrences in the anomalous-event area. Moreover, relatively cold SSTs are also observed by the L3 satellite product (Fig. A3) although only on 7 March and from 14 March onwards (the region is cloudy between 9 and 13 March). Modelling products show that the strong mixing event that started on 9 March and ended on 25 March (Fig. A4) is possibly driven by the cooling, and that the area with the highest mixed layer depths (larger than 1000 m) well overlaps with the April 2022 anomalous bloom. The strong negative heat fluxes into the sea, which occur at the same dates of the cooling events (Fig. A5), further confirm that the driving mechanism of the event is represented by significant heat losses. Considering the Copernicus Marine Ocean Monitoring Indicator (Lyubartsev et al., 2023; product ref. 7, Table 1), in the Levantine basin a large dense water formation rate of approximately 1.3 Sv is documented (not shown[1]) in the winter of 2022. Confirming the relevant effects on physical marine processes of the 2022 cold outbreak, the same LDW formation index was higher only during the noteworthy Eastern Mediterranean transient (EMT; 1992-1993), when the formation rate reached up to 1.8 Sv. After that, the LDW formation index showed only two maxima in 2008 and 2012 with relatively low values (0.7 Sv and 1.0 Sv, respectively).

Slightly later with respect to AST minima and during the strong mixing in March 2022, in the event area southeast of Crete (Fig. 1) the phosphocline depth (PCLD; depth of the maximum phosphate vertical gradient; Salon et al., 2019) is deeper than 400 m and of the PCLD climatology 99$^{th}$ percentile, with two peaks that go down to nearly 600 m (Fig. 2b). The March 2022 anomalous deepening of the phosphocline is preceded in February by a more typical event with deepening of the phosphocline

[1] The Ocean Monitoring Indicator on the water mass formation will be extended to 2022 and published on Copernicus Marine catalogue by 2024.

that stays within its climatology 99th percentile and goes down to nearly 400 m. Due to its role as the limiting nutrient in the Mediterranean Sea (Siokou-Frangou et al., 2010), only phosphocline is included in Fig. 3, but we verified that nitrate is very similarly impacted by the 2022 anomalous event processes.

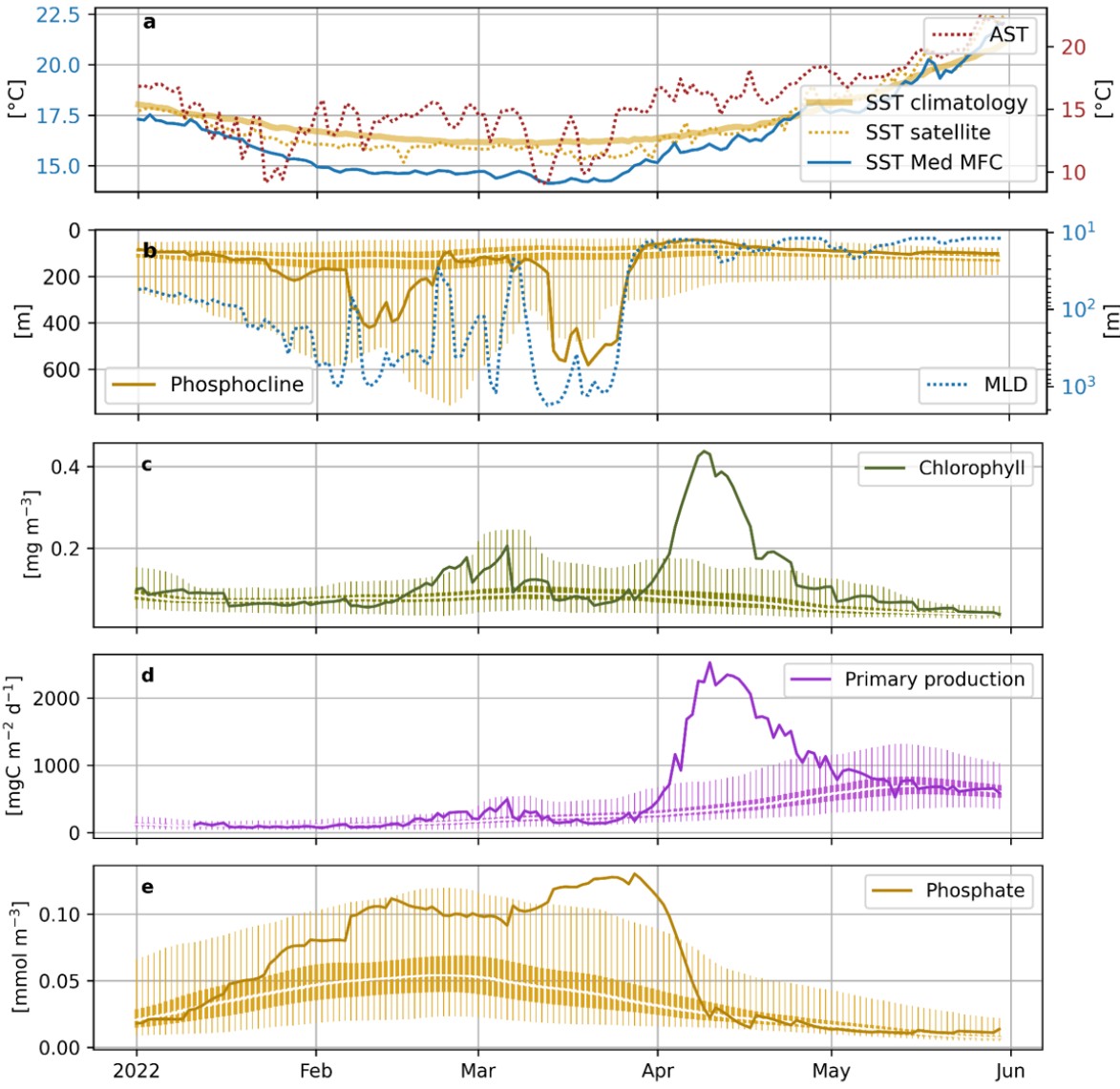

Figure 2: Daily time series - spatially averaged over the event area (Fig. 1) - from January to May 2022 of (a) air surface temperature (AST), sea surface temperature for satellite (SST satellite, product ref. 6, Table 1) and model (SST MED-MFC, product ref. 4, Table 1), and SST satellite climatology; (b) mixed layer depth (MLD: product ref. 4. Table 1) and phosphocline (product ref. 1, Table 1); (c) surface chlorophyll concentration; (d) mean concentration of phosphate above the phosphocline; and (e) primary production integrated in the 0-200 m layer. Climatological percentiles are shown in b, c, d and e (thin vertical line: 1st and 99th percentiles, thick vertical line: 25th and 75th percentiles, white marker: median).

At the bloom peak (8 April) the mean surface chlorophyll concentration in the event area (Feudale et al., 2022; Fig. 1, product

ref. 1, Table 1) is higher than 0.4 mg m$^{-3}$, i.e., more than twice the climatological 99$^{th}$ percentile (Fig. 2c). Chlorophyll
concentrations are higher than the 99$^{th}$ climatological percentile from 2 to 10 April, indicating that the 2022 bloom event is
anomalous both in terms of intensity and timing. In fact, according to the chlorophyll climatology, typical late-winter/early-
spring chlorophyll peaks occur in the first half of March. With similar timing of the anomalous surface chlorophyll
concentration, primary production integrated over the 0-200 m layer largely exceeds the 99$^{th}$ climatological percentile

(Fig. 2d). Related to the strong mixing event occurred in the late winter of 2022 and to the deepening of the phosphocline
(Fig. 2b), the mean phosphate concentration above the phosphocline is higher than the 99$^{th}$ percentile in the month preceding
the phytoplankton bloom with a sharp decrease during the bloom-establishing phase (Fig. 2e). The delay between the large
availability of nutrients in the surface layer and the bloom peak is consistent with the Sverdrup theory (Mayot et al., 2017),
according to which surface bloom starts when the MLD becomes shallower than the euphotic layer. Indeed, when the mixing

is limited to the surface, phytoplankton is no longer diluted over the water column but remains in the surface layer where both
light and nutrients (brought to the surface by the previous deep mixing) are available and favourable to the bloom onset.

The anomalous bloom event is clearly detectable in surface chlorophyll observations (multi-sensor ocean colour product,
Colella et al., 2022; product ref. 3, Table 1) reaching values comparable to those simulated by the Analysis and Forecast system
(Fig. 3). In particular, Fig. 3 shows that high chlorophyll concentrations are observed on 27 and 29 March and on 1 and 6

April, indicating that the event started around 27 March, maintained high concentration values on 29 March, 1 and 6 April and
possibly ended between 8 and 9 April. On these dates, observed chlorophyll concentration is higher than 0.5 mg m$^{-3}$ (up to 3
mg m$^{-3}$ on 29 March). Moreover, high chlorophyll concentrations are located in an area that differs (southwestern shifted) from
the usual "Rhodes gyre" bloom regions, which in Fig. 3 is represented by the magenta contour identifying satellite climatology
above 0.115 mg m$^{-3}$ (i.e., half of the threshold used to define the event area; Fig. 1). Further, we observed that the area with

climatological concentration above the threshold is largest at the beginning of March (not shown). Model daily maps of model
surface chlorophyll concentration provided in the Appendix (Fig. A6) show that in the simulation the bloom started on 4 April,
reached a peak between 8 and 9 April with concentration larger than 0.5 mg m$^{-3}$ (i.e., similar values to the ones observed in
satellite maps), and gradually extinguished from 11 April onwards. An analysis of the deviation of satellite chlorophyll
observations with respect to the 1999-2020 climatology (Fig. A7) demonstrates that on 27 March and 29 March, and on 1 and

8 April chlorophyll is at least 3 standard deviations higher than the mean in the event area.

Since the 2022 anomalous surface bloom is the result of a sequence of processes (cold spell, sea surface cooling, vertical
mixing, fertilisation and subsequent stratification), uncertainties in the representation of each of these dynamics by the
atmospheric-ocean and biogeochemical models may combine and result in inaccuracies in the spatiotemporal representation
of the bloom. However, even if the bloom simulation shows a delay of 5-8 days, the use of three-dimensional modelled data

allowed to: (i) define the temporal and spatial boundaries of the event, and (ii) tackle the sequence of physical and
biogeochemical processes that are involved in the bloom dynamics.

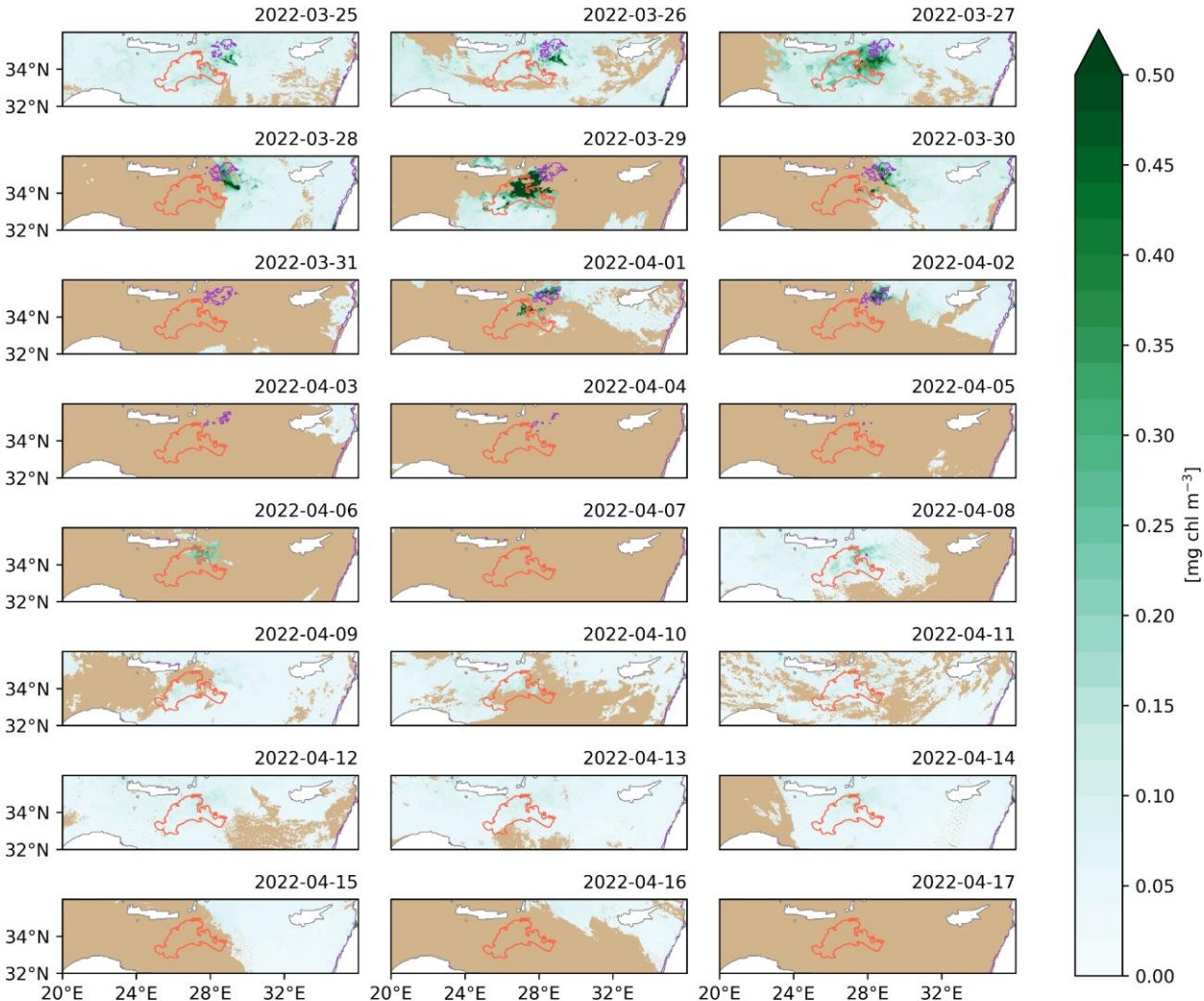

**Figure 3: Daily maps of satellite surface chlorophyll concentration [mg m⁻³] (product ref. 3, Table 1) from 25 March (upper left panel) to 17 April 2022 (lower right panel), orange line contour of the event area (Fig. 1) ), and purple line contour identifying satellite climatology above 0.115 mg m⁻³. Areas without satellite observations are masked in light brown.**

The anomalous localization of the 2022 bloom can be further supported by comparing the vertical processes at two locations (Fig. 4): (i) inside the area of the event and (ii) in the Rhodes gyre area where late winter bloom typically occur ("+" and "x" marker in Fig. 1, respectively). The Hovmöller diagram of temperature inside the event area reveals the gradual outcropping of deep water masses that on 25 March reached the surface from 2000 m (Fig. 4a). At the same time phosphate concentration shows a nearly vertical uniform distribution with persistent high values in the surface layer ($> 0.15$ mmol m⁻³) till the beginning of the event (4 April), when the nutrient started to be consumed (Fig 4c). Starting on 4 April, large chlorophyll concentration in the surface and subsurface layer follows the nutrient injection (Fig. 4e). Finally, a transition to stratified conditions with formation of a deep chlorophyll maximum (DCM) occurs from 10 April. The location outside the 2022 event (right column of

Fig. 4) shows much less intense and shorter water column mixing with lower phosphate concentration in surface layers (Fig. 4b and d). In the chlorophyll Hovmöller diagram (Fig. 4f), a transition phase (non-negligible surface concentration with subsurface chlorophyll maximum; Lavigne et al., 2015) toward summer stratified DCM conditions is already in place at the end of March.

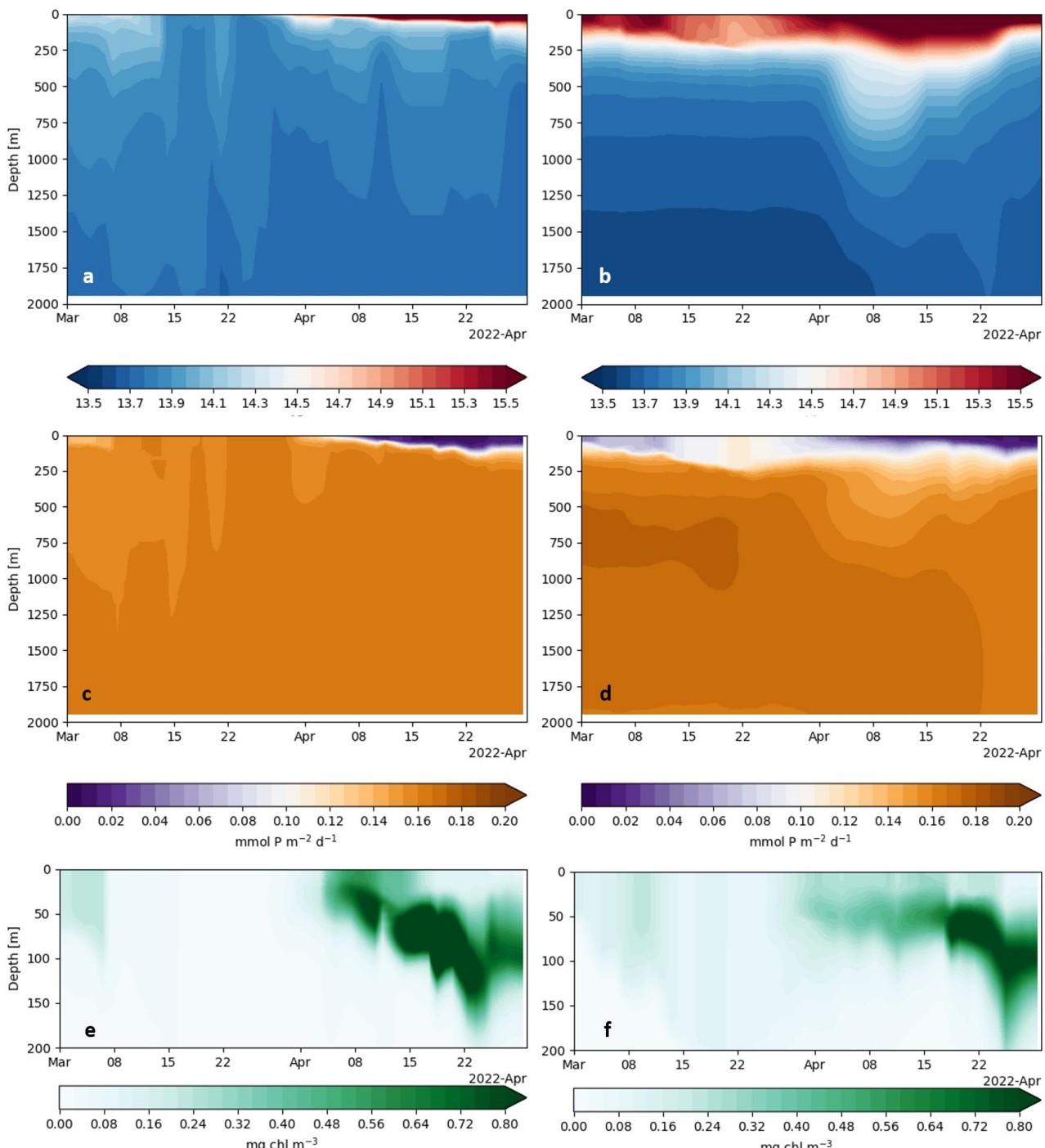

**Figure 4: Hovmöller diagrams in March and April of temperature (top panels; a and b), phosphate and chlorophyll concentrations**
**(middle, c and d, and bottom panels, e and f) inside the event area at 27 °E - 34.02 °N (left panels, white "+" marker in Fig. 1) and**
**outside the event area at 28.5 °E - 35.1 °N (right panels; white "x" marker in Fig. 1).**

# 5 Discussion and conclusions

An anomalous deep mixing and bloom event in the south-eastern Mediterranean in the 2022 late-winter early-spring period was detected by means of the Copernicus Marine MED-MFC products. In this region, intense phytoplankton blooms related to vertical mixing processes and consequent nutrient supply are usually located in the Rhodes gyre area, and have been previously investigated using in-situ and satellite observations and modelling products (e.g., Habib et al., 2022; D'Ortenzio et al., 2021; Varkitzi et al., 2020; Siokou-Frangou et al., 2010), while the 2022 event was located southeast of Crete (Fig. 1). In this work we analysed and described the 2022 event main features and its drivers.

The deep convection and the phytoplankton bloom events in the Cretan area are consistent with the anomalous cooling that occurred in southeastern Europe at the beginning of 2022, showing a dynamics similar to the one of the Mediterranean marine cold spell events described in Simon et al. (2022). For instance, the impact of 2022 cold spells on the North-Central Aegean Sea has been recently demonstrated by Potiris et al. (2024, which show that buoyancy losses during the winter 2021–2022 were comparable to those of 1993–1994, 2002–2003, and 2012, which were all years of dense water formation (DWF) in the Aegean Sea as discussed in Section 3. The findings of Potiris et al. (2024) further support the fact that the 2022 winter and related marine processes can be considered as anomalous for the Eastern Mediterranean. Furthermore, the connection between the 2022 atmospheric conditions and sea cooling is corroborated considering the impacts of atmospheric modes of variability on Mediterranean Sea surface heat exchange discussed by Josey et al. (2011) and Reale et al. (2020). Indeed, both the East Atlantic/Western Russia and the East Atlantic pattern indexes, that are associated with negative heat fluxes in the eastern Mediterranean, were relatively high in March 2022 (Index of /cwlinks, 2023).

The frequency and the impacts of marine extreme events in recent years have been investigated in the Mediterranean Sea (Dayan et al., 2023; Martínez et al., 2023; McAdam et al., 2023; Simon et al., 2022; Darmaraki et al., 2019) also proposing innovative techniques to analyse prolonged episodes in marine ecosystems (Di Biagio et al., 2020). Together with the relatively high number of variables exceeding their 99[th] percentile during the event (Fig. 2 and 3), the recent decrease in the occurrence of cold marine extremes in the Eastern Mediterranean (Simon et al., 2022) further highlights the exceptionality of the 2022 event.

Our study documents the importance of the value chain composed by atmospheric, ocean and biogeochemical prediction models in detecting anomalies with respect to the typical state and variability. In particular, the strong anomaly in phytoplankton bloom intensity revealed to be a suitable descriptor to define the 2022 event localization, extent and duration. Moreover, the evaluation of the spatial and temporal mismatch of the simulated event with respect to the Copernicus Marine ocean colour product provides an assessment of the capability of the prediction chain to simulate specific events.

Considering that previous anomalous cooling events (1992-1993) were among the drivers of the EMT that impacted the whole Mediterranean Sea dynamics (e.g., Pinardi et al., 2019; Roether et al., 2007; Theocharis et al., 2002) with consequences on other marine compartments (e.g., nutrients and productivity, biodiversity, and acidification; Tsiaras et al., 2012; Touratier and

Goyet, 2011; Danovaro et al., 2004; Stratford and Haines, 2002; Civitarese and Gacic, 2001), the 2022 event and the related deep water formation might be worth further investigation.

Our results show that the 2022 anomalous event increased by 35% the annual primary production in an area of approximately 35000 km$^2$ (i.e., 11% and 1.4% of the Levantine basin and Mediterranean Sea surface, respectively). As a consequence of the increased organic matter synthesis, a non-negligible impact along the whole food chain might have occurred, given the well proven link between productivity and fish catches (Canu et al., 2022; Colloca et al., 2017; Piroddi et al., 2017; Conti and Scardi, 2010). Due to the relative time proximity of the event and to the non-trivial work needed to collect fishing catches data, it was not possible to gather quantitative information on this aspect, although the impact of the 2022 anomaly on the higher trophic level deserves a closer look.

**Appendix A: Daily maps of observed and simulated marine properties in the Cretan area**

The spatial and temporal extent of the 2022 bloom event and of its drivers can be further investigated using daily maps of a range of marine features in the Cretan area during March 2022. Atmospheric, model sea surface temperature anomaly and satellite sea surface temperature are shown in Figs. A1-A3, providing evidence of the intense cooling event that occurred in March 2022, and described in the Results section. Looking at the MLD daily maps (Fig. A4), it is worth noting that on 3 and 4 March the bloom event is already interested by deep mixing (also visible in Fig. 2), possibly related to the relatively low SST values already present at the beginning of March (Fig. A2) and resulting from early-winter cooling events (starting in January 2022, as shown by the AST time series in Fig. 2). Remarkably, the strong MLD deepenings in the anomalous event area are followed by positive total heat fluxes in the sea (e.g., 17 March, Fig. A5), due to the deep water reaching the surface layers together with its temperature being lower than AST.

Finally, the spatio-temporal evolution of the event as simulated by the Mediterranean Sea Copernicus Marine forecasting centre is detailed in model daily maps of surface chlorophyll concentration in the same dates of satellite observations provided in Fig. 4 (from 25 March to 17 April 2022, Fig. A6).

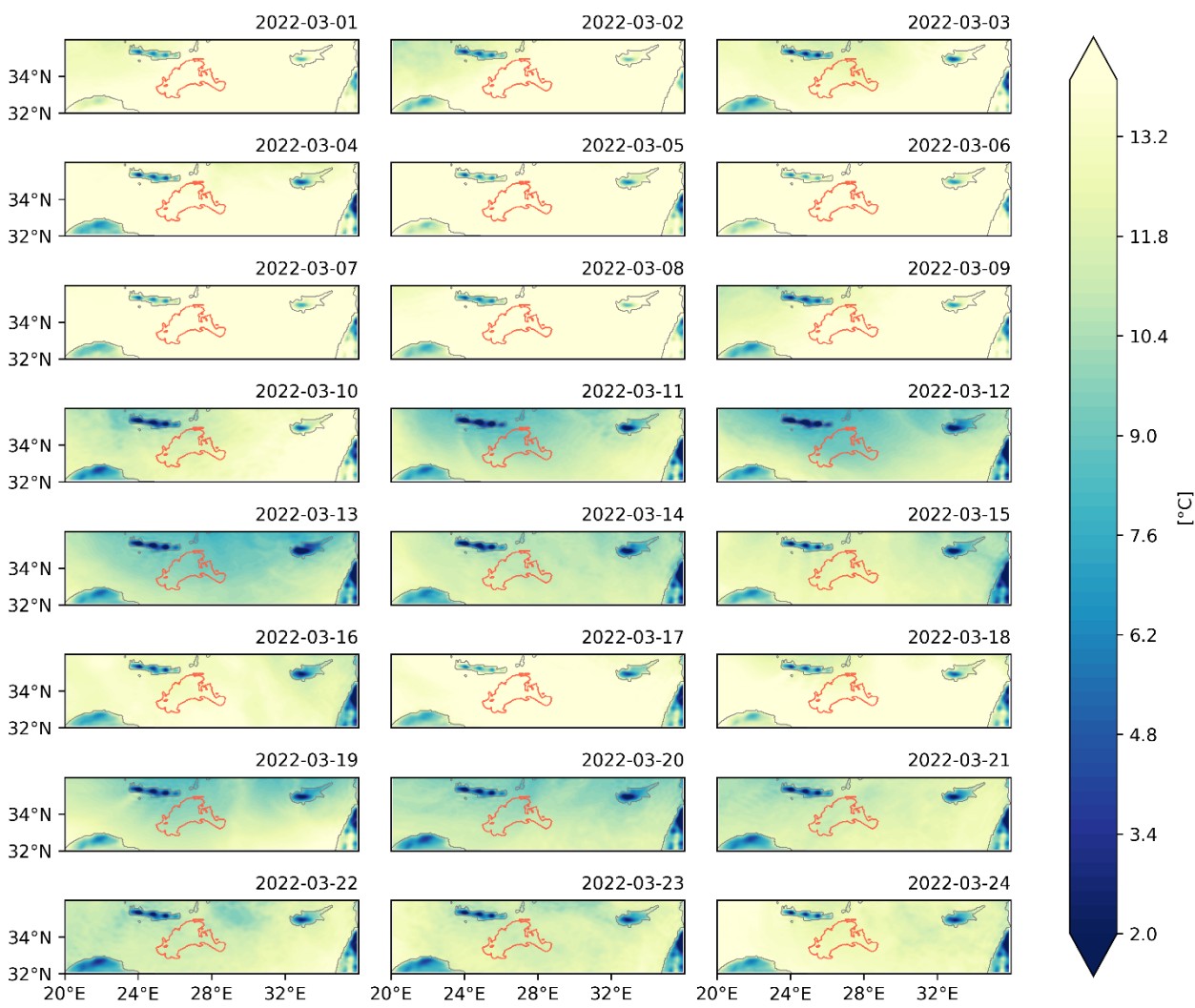

**Figure A1: Daily maps of ECMWF atmospheric surface temperature [°C] (product ref. 8, Table 1) from 1 March (upper left panel) to 24 March 2022 (lower right panel), and orange line contour of the event area (Fig. 1).**

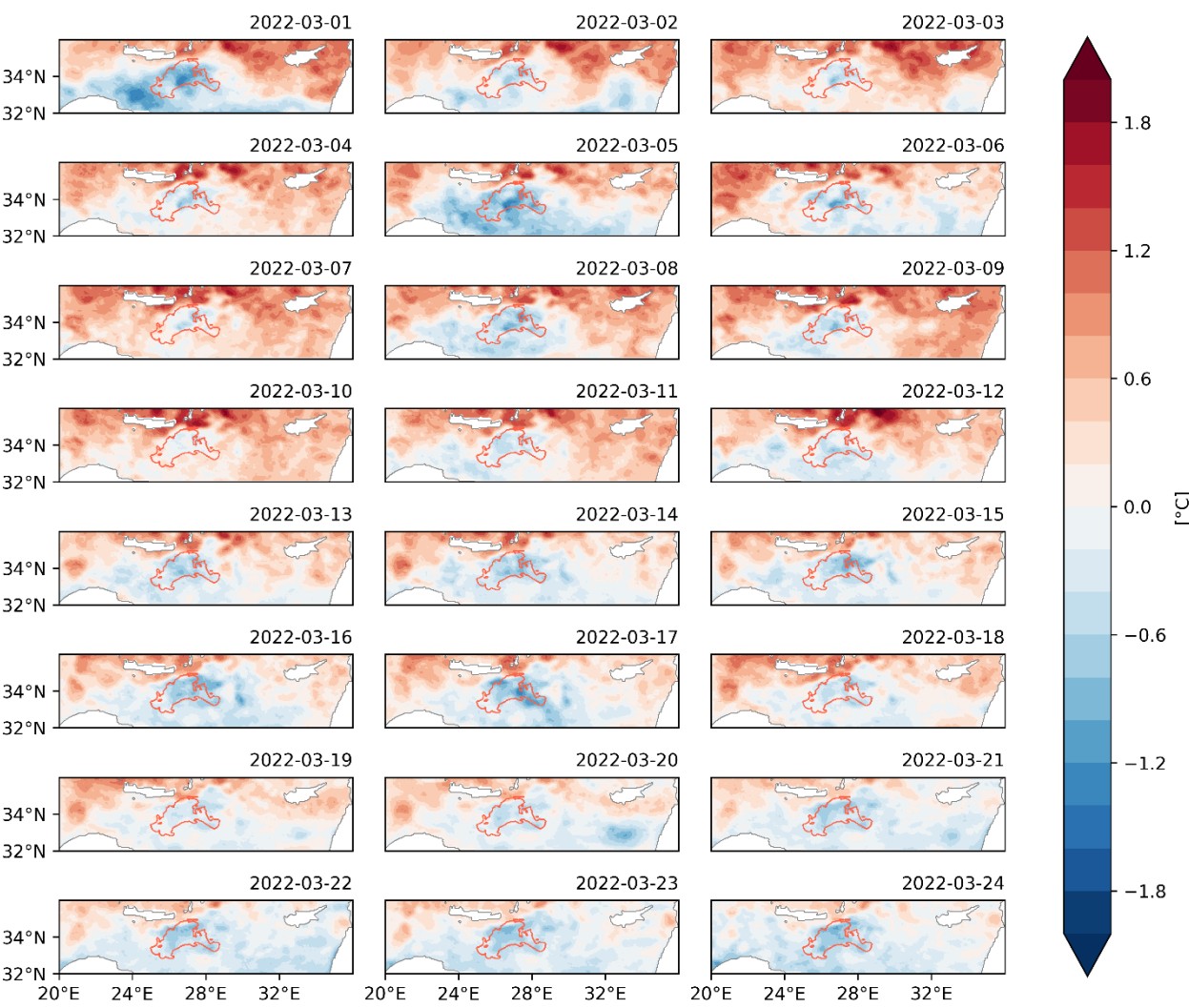

**Figure A2: Daily maps of model sea surface temperature anomaly [°C] (product ref. 4, Table 1) from 1 March (upper left panel) to 24 March 2022 (lower right panel), and orange line contour of the event area (Fig. 1).**

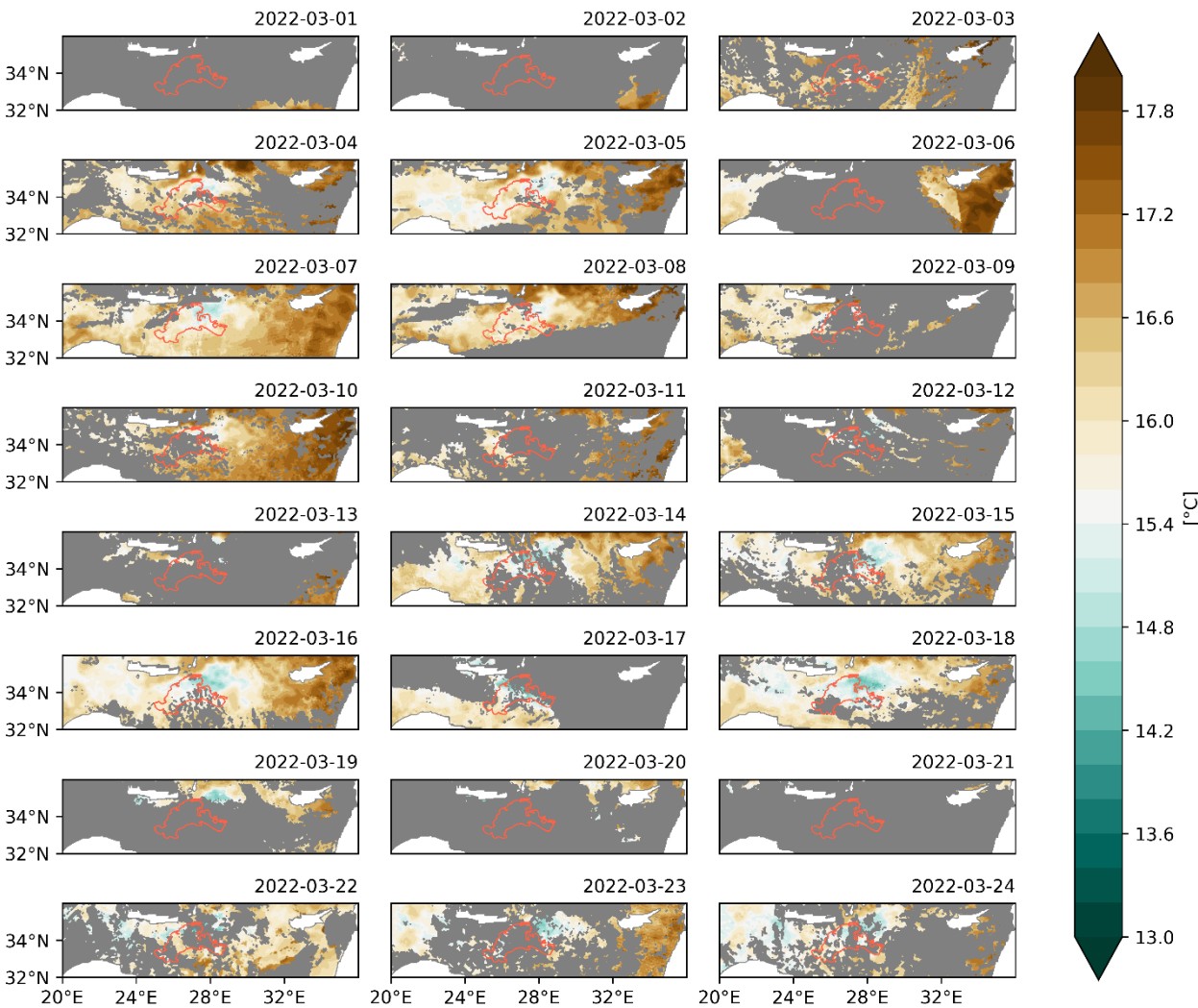

**Figure A3: Daily maps of satellite sea surface temperature [°C] (product ref. 6, Table 1) from 1 March (upper left panel) to 24 March 2022 (lower right panel), and orange line contour of the event area (Fig. 1). Areas without satellite observations are masked with grey.**

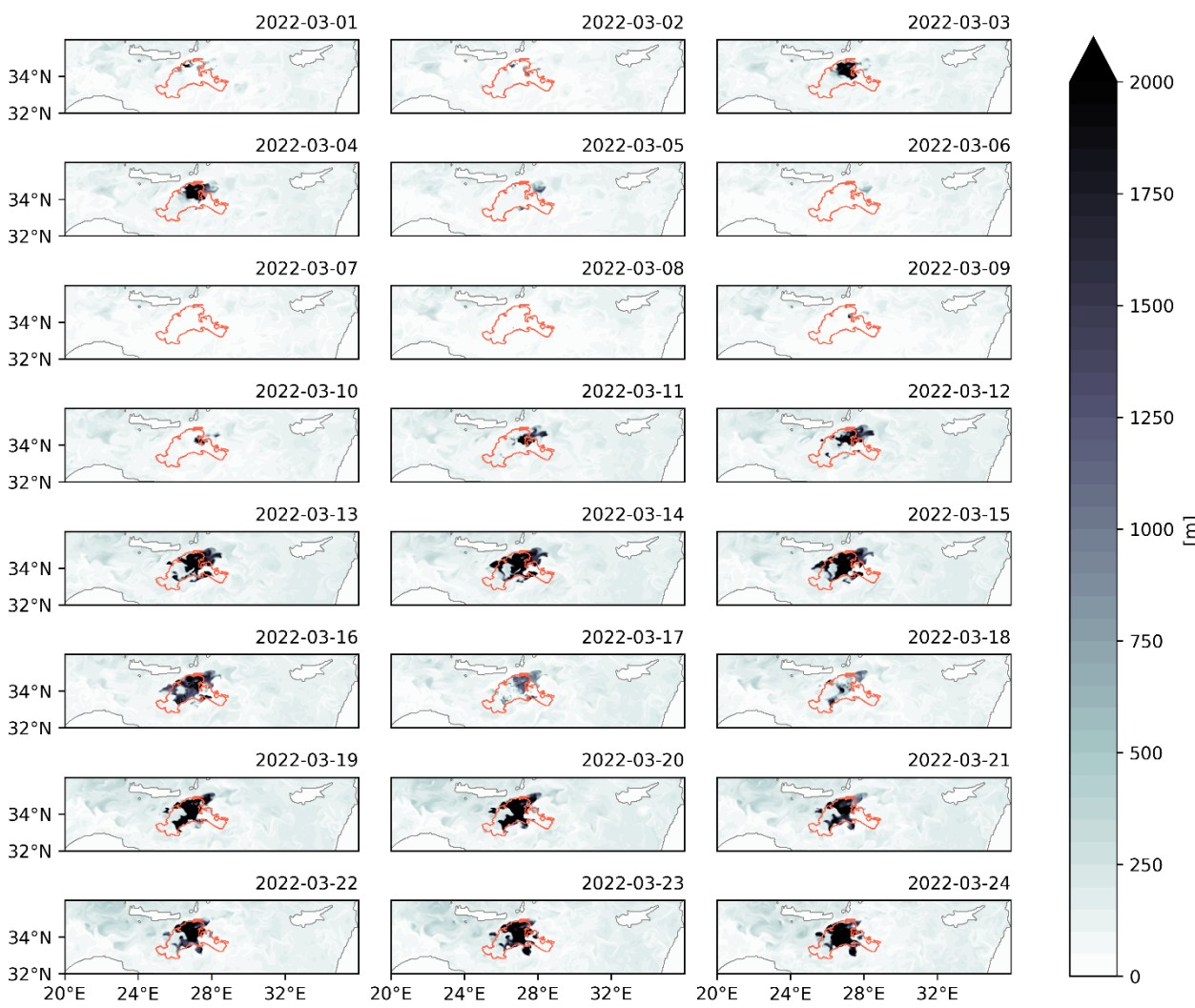

**Figure A4: Daily maps of mixed layer depth [m] (product ref. 4. Table 1) from 1 March (upper left panel) to 24 March 2022 (lower right panel), and orange line contour of the event area (Fig. 1).**

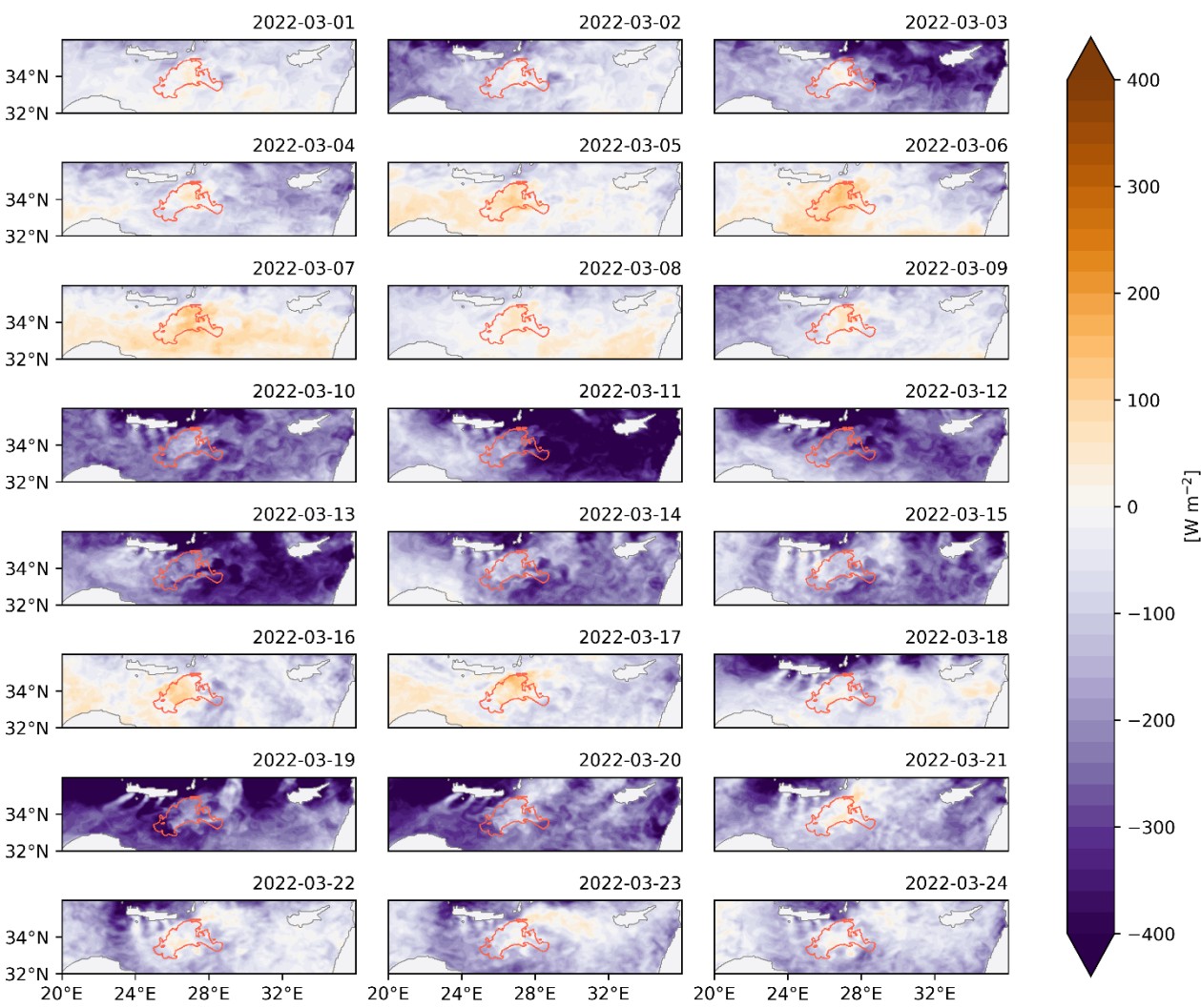

**Figure A5: Daily maps of total heat fluxes in sea water [W m$^{-2}$] (product ref. 4. Table 1) from 1 March (upper left panel) to 24 March 2022 (lower right panel), and orange line contour of the event area (Fig. 1).**

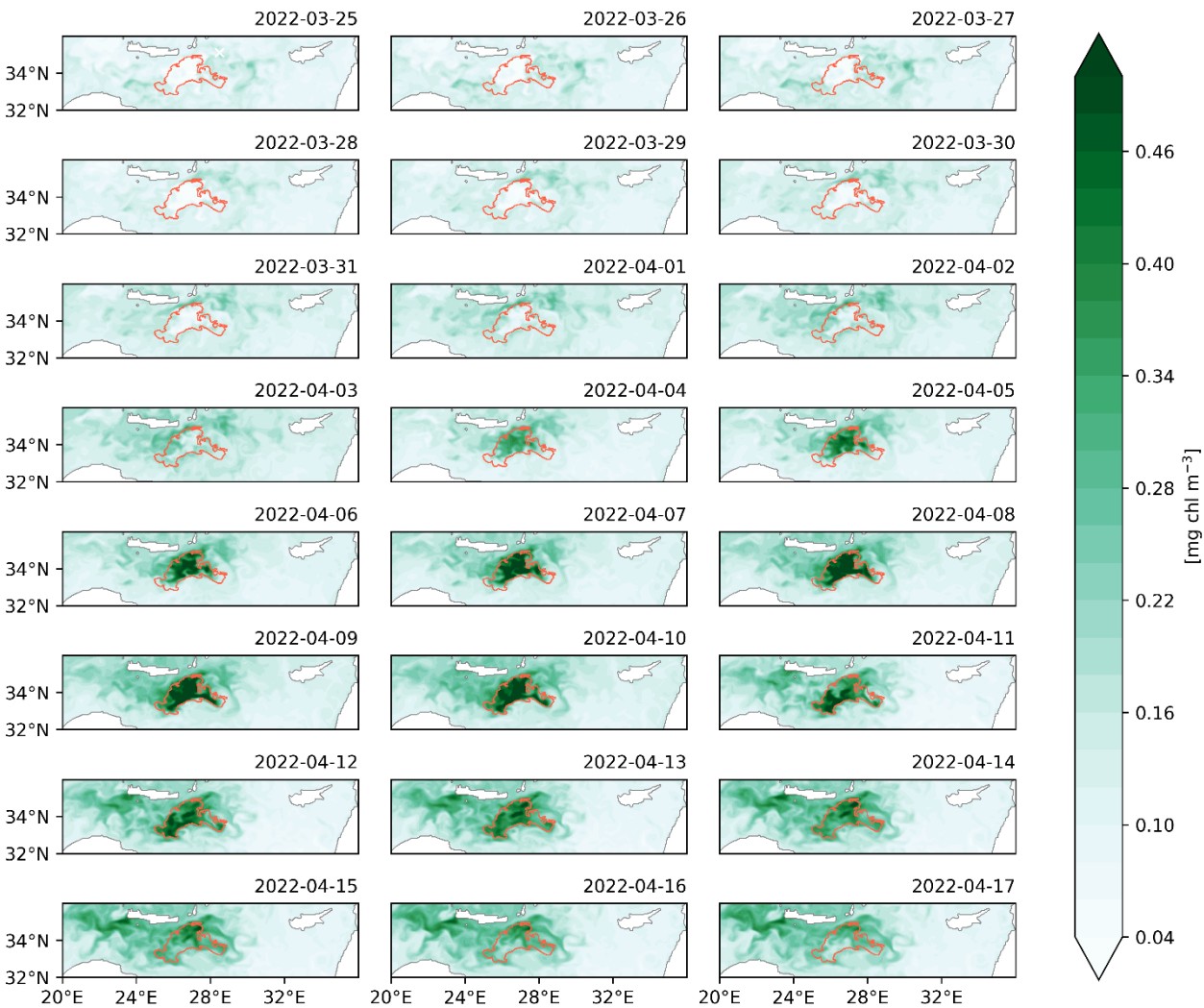

**Figure A6: Daily maps of mode surface chlorophyll concentration [mg m$^{-3}$] (product ref. 1, Table 1) from 25 March (upper left panel) to 17 April 2022 (lower right panel), and orange line contour of the event area (Fig. 1).**

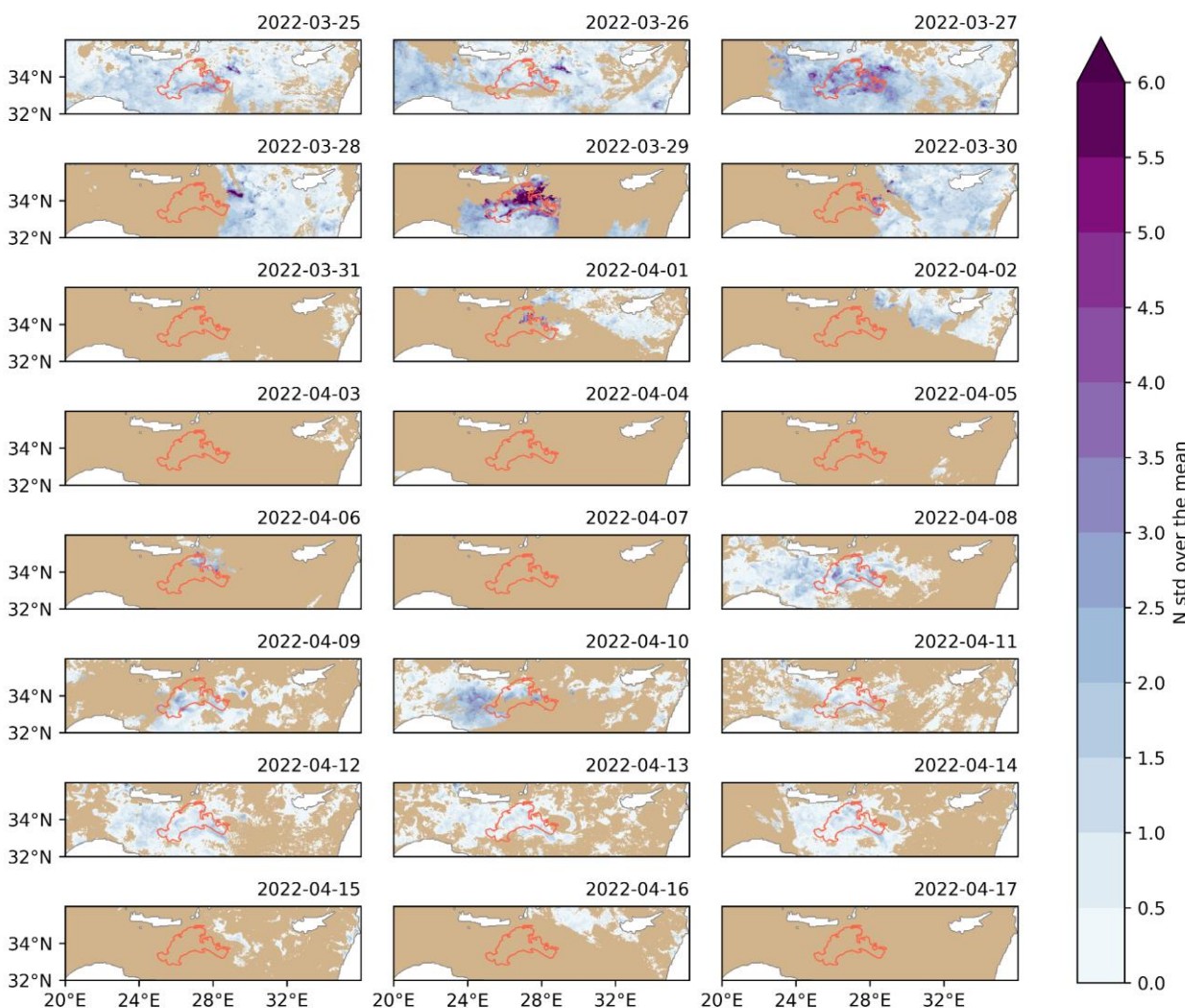

**Figure A7: Daily maps of number of standard deviations over the climatological mean (product ref. 3, Table 1) from 25 March (upper left panel) to 17 April 2022 (lower right panel), and orange line contour of the event area (Fig. 1).. Areas without satellite observations are masked with light brown.**

**Data availability**

Publicly available datasets were analysed in this study. Modelling and in situ data can be found at the Copernicus Marine Service, with references and DOIs indicated in the Table 1 of the manuscript.

## Author contribution

AT, AA and GC conceived the idea. AT, AA, CF and SC conducted the analysis. AT, AA and GC wrote the first draft, with contributions from the other co-authors. All the authors discussed and reviewed the submitted manuscript.

## Competing interests

The authors declare that they have no conflict of interest.

## Acknowledgements

This study has been conducted using EU Copernicus Marine Service Information.

## Financial support

This research has been partly supported by the MED-MFC "Mediterranean Monitoring and Forecasting Centre" of Copernicus Marine Service, which is implemented by Mercator Ocean International within the framework of a delegation agreement with the European Union. (Ref.n.21002L5-COP-MFCMED-5500).

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

Index of /cwlinks: https://ftp.cpc.ncep.noaa.gov/cwlinks/, last access: 31 July 2023.

Josey, S. A., Somot, S., and Tsimplis, M.: Impacts of atmospheric modes of variability on Mediterranean Sea surface heat exchange, Journal of Geophysical Research: Oceans, 116, https://doi.org/10.1029/2010JC006685, 2011.

Kubin, E., Poulain, P.-M., Mauri, E., Menna, M., and Notarstefano, G.: Levantine Intermediate and Levantine Deep Water Formation: An Argo Float Study from 2001 to 2017, Water, 11, 1781, https://doi.org/10.3390/w11091781, 2019.

Lazzari, P., Solidoro, C., Ibello, V., Salon, S., Teruzzi, A., Béranger, K., Colella, S., and Crise, A.: Seasonal and inter-annual variability of plankton chlorophyll and primary production in the Mediterranean Sea: a modelling approach, Biogeosciences, 9, 217–233, https://doi.org/10.5194/bg-9-217-2012, 2012.

Lecci, R., Salon S., Bolzon, G., and Cossarini, G.: EU Copernicus Marine Service Product User Manual for the Mediterranean Sea Biogeochemistry Analysis and Forecast, MEDSEA_ANALYSISFORECAST_BGC_006_014, Issue 2.2, Mercator Ocean International, https://catalogue.marine.copernicus.eu/documents/PUM/CMEMS-MED-PUM-006-014.pdf, last access: 20 June 2023, 2022a.

Lecci, R., Salon S., Bolzon, G., and Cossarini, G.: EU Copernicus Marine Service Product User Manual for the Mediterranean Sea Biogeochemistry Reanalysis, MEDSEA_MULTIYEAR_BGC_006_008, Issue 3.2, Mercator Ocean International, https://catalogue.marine.copernicus.eu/documents/PUM/CMEMS-MED-PUM-006-008.pdf, last access: 20 June 2023, 2022b.

Lecci, R., Drudi, M., Grandi, A., Cretì, S., and Clementi, E.: EU Copernicus Marine Service Product User Manual for the the Mediterranean Sea Physics Analysis and Forecast, MEDSEA_ANALYSISFORECAST_PHY_006_013, Issue 2.2, Mercator Ocean International, https://catalogue.marine.copernicus.eu/documents/PUM/CMEMS-MED-PUM-006-013.pdf, last access: 20 June 2023, 2022c.

Lecci, R., Drudi, M., Grandi, A., Cretì, S., and Clementi, E.: EU Copernicus Marine Service Product User Manual forMediterranean Sea Physics Reanalysis, , MEDSEA_MULTIYEAR_PHY_006_004, https://catalogue.marine.copernicus.eu/documents/PUM/CMEMS-MED-PUM-006-004.pdf, Issue 2.3, Mercator Ocean International,, last access: 17 July 2023, 2022d.

Lyubartsev, V., and Clementi, E.: EU Copernicus Marine Service Product User Manual for the Mediterranean Water Mass Formation Rates from Reanalysis, OMI_VAR_EXTREME_WMF_MEDSEA_area_averaged_mean, Issue 1.0, Mercator Ocean International, https://catalogue.marine.copernicus.eu/documents/PUM/CMEMS-OMI-PUM-VAR-EXTREME-WMF-MEDSEA.pdf, last access: 20 June 2023, 2022.

Lyubartsev, V., Clementi, E., Aydogdu, A., Miraglio, P., Masina, S., and Pinardi, N.: EU Copernicus Marine Service Quality Information Document for the Mediterranean Water Mass Formation Rates from Reanalysis, OMI_VAR_EXTREME_WMF_MEDSEA_area_averaged_mean, Issue 1.0, Mercator Ocean International, https://catalogue.marine.copernicus.eu/documents/QUID/CMEMS-OMI-QUID-VAR-EXTREME-WMF-MEDSEA.pdf, last access: 20 June 2023, 2023.

Martínez, J., Leonelli, F. E., García-Ladona, E., Garrabou, J., Kersting, D. K., Bensoussan, N., and Pisano, A.: Evolution of marine heatwaves in warming seas: the Mediterranean Sea case study, Frontiers in Marine Science, 10, https://doi.org/10.3389/fmars.2023.1193164, 2023.

Mayot, N., D'Ortenzio, F., Taillandier, V., Prieur, L., Fommervault, O. P. de, Claustre, H., Bosse, A., Testor, P., and Conan, P.: Physical and Biogeochemical Controls of the Phytoplankton Blooms in North Western Mediterranean Sea: A Multiplatform Approach Over a Complete Annual Cycle (2012–2013 DEWEX Experiment), Journal of Geophysical Research: Oceans, 122, 410 9999–10019, https://doi.org/10.1002/2016JC012052, 2017.

McAdam, R., Masina, S., and Gualdi, S.: Seasonal forecasting of subsurface marine heatwaves, Commun Earth Environ, 4, 1–11, https://doi.org/10.1038/s43247-023-00892-5, 2023.

Pinardi, N., Cessi, P., Borile, F., and Wolfe, C. L. P.: The Mediterranean Sea Overturning Circulation, Journal of Physical Oceanography, 49, 1699–1721, https://doi.org/10.1175/JPO-D-18-0254.1, 2019.

Piroddi, C., Coll, M., Liquete, C., Macias, D., Greer, K., Buszowski, J., Steenbeek, J., Danovaro, R., and Christensen, V.: Historical changes of the Mediterranean Sea ecosystem: modelling the role and impact of primary productivity and fisheries changes over time, Sci Rep, 7, 44491, https://doi.org/10.1038/srep44491, 2017.

Pisano, A., Fanelli, C., Cesarini, C., Tronconi, C., La Padula, F., and Buongiorno Nardelli, B.: EU Copernicus Marine Service Quality Information Document for the Mediterranean Sea High Resolution L4 Sea Surface Temperature Reprocessed, 420 SST_MED_SST_L4_REP_OBSERVATIONS_010_021, Issue 2.0, Mercator Ocean International, https://catalogue.marine.copernicus.eu/documents/QUID/CMEMS-SST-QUID-010-021-022-041-042.pdf, last access: 20 June 2023, 2022a.

Pisano, A., Fanelli, C., Cesarini, C., Tronconi, C., La Padula, F., and Buongiorno Nardelli, B.: EU Copernicus Marine Service Product User Manual for the Mediterranean Sea High Resolution L4 Sea Surface Temperature Reprocessed, 425 SST_MED_SST_L4_REP_OBSERVATIONS_010_021, Issue 2.0, Mercator Ocean International, https://catalogue.marine.copernicus.eu/documents/PUM/CMEMS-SST-PUM-010-021-022-041-042.pdf, last access: 20 June 2023, 2022b.

Potiris, M., Mamoutos, I. G., Tragou, E., Zervakis, V., Kassis, D., and Ballas, D.: Dense Water Formation in the North–Central Aegean Sea during Winter 2021–2022, Journal of Marine Science and Engineering, 12, 221, 430 https://doi.org/10.3390/jmse12020221, 2024.

Reale, M., Salon, S., Somot, S., Solidoro, C., Giorgi, F., Crise, A., Cossarini, G., Lazzari, P., and Sevault, F.: Influence of large-scale atmospheric circulation patterns on nutrient dynamics in the Mediterranean Sea in the extended winter season (October-March) 1961-1999, Climate Research, 82, 117–136, https://doi.org/10.3354/cr01620, 2020.

Roether, W., Klein, B., Manca, B. B., Theocharis, A., and Kioroglou, S.: Transient Eastern Mediterranean deep waters in 435 response to the massive dense-water output of the Aegean Sea in the 1990s, Progress in Oceanography, 74, 540–571, https://doi.org/10.1016/j.pocean.2007.03.001, 2007.

Salon, S., Cossarini, G., Bolzon, G., Feudale, L., Lazzari, P., Teruzzi, A., Solidoro, C., and Crise, A.: Novel metrics based on Biogeochemical Argo data to improve the model uncertainty evaluation of the CMEMS Mediterranean marine ecosystem forecasts, Ocean Science, 15, 997–1022, https://doi.org/10.5194/os-15-997-2019, 2019.

Simon, A., Plecha, S. M., Russo, A., Teles-Machado, A., Donat, M. G., Auger, P.-A., and Trigo, R. M.: Hot and cold marine extreme events in the Mediterranean over the period 1982-2021, Frontiers in Marine Science, 9, 2022.

Siokou-Frangou, I., Christaki, U., Mazzocchi, M. G., Montresor, M., Ribera d'Alcalá, M., Vaqué, D., and Zingone, A.: Plankton in the open Mediterranean Sea: a review, Biogeosciences, 7, 1543–1586, https://doi.org/10.5194/bg-7-1543-2010, 2010.

Stratford, K. and Haines, K.: Modelling nutrient cycling during the eastern Mediterranean transient event 1987–1995 and beyond, Geophysical Research Letters, 29, 5-1-5–4, https://doi.org/10.1029/2001GL013559, 2002.

Surface air temperature for March 2022 | Copernicus: https://climate.copernicus.eu/surface-air-temperature-march-2022, last access: 28 July 2023.

Teruzzi, A., Di Biagio, V., Feudale, L., Bolzon, G., Lazzari, P., Salon, S., Coidessa, G., and Cossarini, G.: EU Copernicus Marine Service Quality Information Document for the Mediterranean Sea Biogeochemistry Reanalysis, MEDSEA_MULTIYEAR_BGC_006_008, Issue 3.2, Mercator Ocean International, https://catalogue.marine.copernicus.eu/documents/QUID/CMEMS-MED-QUID-006-008.pdf, last access: 20 June 2023, 2022.

Theocharis, A., Klein, B., Nittis, K., and Roether, W.: Evolution and status of the Eastern Mediterranean Transient (1997–1999), Journal of Marine Systems, 33–34, 91–116, https://doi.org/10.1016/S0924-7963(02)00054-4, 2002.

Touratier, F. and Goyet, C.: Impact of the Eastern Mediterranean Transient on the distribution of anthropogenic CO2 and first estimate of acidification for the Mediterranean Sea, Deep Sea Research Part I: Oceanographic Research Papers, 58, 1–15, https://doi.org/10.1016/j.dsr.2010.10.002, 2011.

Tsiaras, K. P., Kourafalou, V. H., Raitsos, D. E., Triantafyllou, G., Petihakis, G., and Korres, G.: Inter-annual productivity variability in the North Aegean Sea: Influence of thermohaline circulation during the Eastern Mediterranean Transient, Journal of Marine Systems, 96–97, 72–81, https://doi.org/10.1016/j.jmarsys.2012.02.003, 2012.

Varkitzi, I., Psarra, S., Assimakopoulou, G., Pavlidou, A., Krasakopoulou, E., Velaoras, D., Papathanassiou, E., and Pagou, K.: Phytoplankton dynamics and bloom formation in the oligotrophic Eastern Mediterranean: Field studies in the Aegean, Levantine and Ionian seas, Deep Sea Research Part II: Topical Studies in Oceanography, 171, 104662, https://doi.org/10.1016/j.dsr2.2019.104662, 2020.