# Peer review of "Anomalous 2022 deep water formation and intense phytoplankton bloom in the Cretan area"

_State of the Planet, 2023_

## Referee Comment (RC1)

Referee Report, Manuscript Number: sp-2023-30

*Anomalous 2022 deep water formation and intense phytoplankton bloom in the Cretan area* by Anna Teruzzi, Ali Aydogdu, Carolina Amadio, Emanuela Clementi, Simone Colella, Valeria Di Biagio, Massimiliano Drudi, Claudia Fanelli, Laura Feudale, Alessandro Grandi, Pietro Miraglio, Andrea Pisano, Jenny Pistoia, Marco Reale, Stefano Salon, Gianluca Volpe, Gianpiero Cossarini

The paper provides evidence of a phytoplankton bloom South-East of Crete island, Eastern Mediterranean Sea, whose location is displaced from the Rhodes gyre, where phytoplankton blooms have been frequently observed in the past. The authors propose that the bloom was triggered by strong vertical mixing events due a cold spell, which brought nutrient in the photic zone, followed by water column stratification. In this respect the succession of events would perfectly match the Sverdrup conceptual model. The authors limit their analysis to the description of the satellite observations and the results of model simulations. This is likely linked to the scope of the issue to which the paper has been submitted.

The pro of the contribution is that it is a good example of how the Copernicus products may be integrated to detect and describe ocean dynamics. Because of this it may contribute to the SP issue.

My perplexities about publishing the paper in its present format are the following.

1. The satellite coverage is quite coarse in time due to cloud coverage (see fig. 4 in the text) and the bloom area reported in fig. 1 which, if I understood well, is produced by the numerical model, does not seem to be supported by the observations, both in space and in time. This may question the estimates of the bloom relevance which, I assume, is based on the model.
2. The authors highlight that the location of the bloom is not the Rhodes gyre, where the cyclonic circulation and the convection often trigger phytoplankton accumulation. Indeed in their map on fig. 1 the South-West border of Rhodes gyre displays low biomass. One then wonders which 3-D dynamics was active so to produce a localized bloom. If the forcing was the strong negative heat flux, this should have acted over the whole area. Why the bloom occurred only in that limited area and there was no bloom in the Rhodes area. Having the model simulations for the whole basin the authors should discuss this aspect.
3. In Fig. 23 the authors show the time course phosphate concentration above the nutricline. Why phosphate? Because is considered the limiting nutrient? One wonders if the nutricline was relative to a specific nutrient or all the nutrient profiles overlapped.
4. There is a time mismatch between satellite and model. The authors acknowledge this, if it is not a mistype, on lines 175-176. However I do not understand why this " *...provides an assessment of the capability of the prediction chain to simulate specific events*". Do the author mean that the assessment suggests that the model did not simulate the event correctly? If so why they are mostly relying the simulations in discussing the event? A clarification would help.
5. Assuming that the model simulation captured to a reasonable extent the dynamics of the mixed layer, Fig. 2 shows that during the bloom time there were three, if not four, events of deep remixing of the water column. This questions the simple reconstruction of the bloom as convection-nutrient upward transport-surface stabilization. Could the authors analyze the dynamics in more detail?
6. The authors mention that "*..the local fishery community reported increased catches..*" but they do not say where and when. The bloom is quite far from the coast. Did the catches increase in the high sea?

The reference Josey, S. and Schroeder, K.: https://doi.org/10.5194/egusphere-egu23-5884, 2023 is never cited in the text.

---

## Author Response (AR1)

Author point-by-point responses to Reviewers comments for the manuscript

**"Anomalous 2022 deep water formation and intense phytoplankton bloom in the Cretan area"**

March, 2024

**Reviewer #1**

**The paper provides evidence of a phytoplankton bloom South-East of Crete island, Eastern Mediterranean Sea, whose location is displaced from the Rhodes gyre, where phytoplankton blooms have been frequently observed in the past. The authors propose that the bloom was triggered by strong vertical mixing events due a cold spell, which brought nutrient in the photic zone, followed by water column stratification. In this respect the succession of events would perfectly match the Sverdrup conceptual model. The authors limit their analysis to the description of the satellite observations and the results of model simulations. This is likely linked to the scope of the issue to which the paper has been submitted.**

**The pro of the contribution is that it is a good example of how the Copernicus products may be integrated to detect and describe ocean dynamics. Because of this it may contribute to the SP issue.**

We thank the Reviewer #1 for the constructive comments on our manuscript. Our point-by-point responses together with propositions for manuscript updates (in green) are provided below. Figures are inserted at the end of the document while line numbers refer to the new version of the manuscript..

**My perplexities about publishing the paper in its present format are the following.**

**1. The satellite coverage is quite coarse in time due to cloud coverage (see fig. 4 in the text) and the bloom area reported in fig. 1 which, if I understood well, is produced by the numerical model, does not seem to be supported by the observations, both in space and in time. This may question the estimates of the bloom relevance which, I assume, is based on the model.**

We agree with the Reviewer that ocean colour (OC) observations (Fig. 4 in the manuscript) are incomplete to spotlight the full extent of the event. However, the comparison of the available observations with the OC climatology (Fig. R1 and R2) reveals higher-than-usual chlorophyll concentrations as well as an intense bloom that is spatially and temporally shifted with respect to the usual patterns observed in the Cretan area.

In particular, Fig. R1 shows that high chlorophyll concentrations are observed on 27 and 29 March and on 1 and 6 April. On these dates, observed chlorophyll concentration is higher than 0.5 mg m$^{-3}$ (up to 3 mg m$^{-3}$ on 29 March). Moreover, high chlorophyll concentrations are located in an area that differs (southwestern shifted) from the usual "Rhodes gyre" bloom

regions, which in Fig. R1 is represented by the magenta contour identifying satellite climatology above 0.115 mg m$^{-3}$ (half of the threshold used to define the event area; Fig 1 in the submitted manuscript). Further, it is worth noting that the area with climatological concentration above the threshold is largest at the beginning of March.

Additionally, an analysis of the deviation of OC observations with respect to the 1999-2020 climatology in the area under investigation demonstrates that on 27 March and 29 March, and on 1 and 6 april chlorophyll is 4 standard deviations higher than the mean (Fig. R2).

For sake of clarity and to include further comparison with observations, daily maps of model surface chlorophyll concentration are provided in Fig. R3. The simulated bloom started on 4 April, reached a peak between 8 and 9 April with concentration larger than 0.5 mg/m3 (i.e., similar values to the ones observed in satellite maps), and gradually extinguished from 11 April onward. On the other hand, from the analysis of satellite maps, it can be presumed that the event started somewhen around 27 March, maintained high concentration values on 29 March, 1 and 6 April and possibly ended between 8 and 9 April.

Even if the simulation shows a delay of 5-8 days, the use of modelled data have some clear advantages since the 3D products allowed to: (i) define the temporal and spatial boundaries of the event, (ii) tackle the sequence of physical and biogeochemical processes that are involved in the bloom dynamics. Indeed, as reported in the following answers, we demonstrate how the bloom started after cold spell events followed by anomalous mixing and subsequent fertilisation of the photic zone.

We believe that our contribution is a good example of the capability of an operational analysis and forecasting system to predict marine anomalous events and to provide a consistent and coherent picture of the processes involved using multiple information sources: atmospheric data, marine physical and biogeochemical modelling results, and observations.

Given the limits imposed by the State of the Planet Journal to the number of figures, we propose the following changes in the new version of the manuscript:

- Substitute old Fig. 4 with Fig. R1b to show the high chlorophyll values observed in OC and their temporal and spatial shift with respect to the area typically impacted by the Rhode gyre bloom.
- Include Fig. R2b and R3 in the novel Appendix A.
- Enrich the result section with a paragraph that highlights how the observations provide evidence of an anomalous event in the area as illustrated above: and the explanation of the observed and simulated sequence of the event as supplementary material in the Results section: "L149 - In particular, Fig. 3 shows that high chlorophyll concentrations are observed on 27 and 29 March and on 1 and 6 April, indicating that the event started around 27 March, maintained high concentration values on 29 March, 1 and 6 April and possibly ended between 8 and 9 April. On these dates, observed chlorophyll concentration is higher than 0.5 mg m-3 (up to 3 mg m-3 on 29 March). Moreover, high chlorophyll concentrations are located in an area that differs (southwestern shifted) from the usual "Rhodes gyre" bloom regions, which in Fig. 3 is represented by the magenta contour identifying satellite climatology above 0.115 mg m-3 (i.e., half of the threshold used to define the event area; Fig. 1).

Further, we observed that the area with climatological concentration above the threshold is largest at the beginning of March (not shown). Model daily maps of model surface chlorophyll concentration provided in the Appendix (Fig. A6) show that in the simulation the bloom started on 4 April, reached a peak between 8 and 9 April with concentration larger than 0.5 mg m-3 (i.e., similar values to the ones observed in satellite maps), and gradually extinguished from 11 April onwards. An analysis of the deviation of satellite chlorophyll observations with respect to the 1999-2020 climatology (Fig. A7) demonstrates that on 27 March and 29 March, and on 1 and 8 April chlorophyll is at least 3 standard deviations higher than the mean in the event area."

- Enrich the discussion section on the mechanism driving late-winter/early-spring blooms in the Levantine basin also considering experimental and modelling studies (e.g., Habib et al., 2023; D'Ortenzio et al., 2021): "L 188 - An anomalous deep mixing and bloom event in the south-eastern Mediterranean in the 2022 late-winter early early-spring period was detected by means of the Copernicus Marine MEDMed-MFC products. In this region, intense phytoplankton blooms related to vertical mixing processes and consequent nutrient supply are usually located in the Rhodes gyre area, and have been previously investigated using in-situ and satellite observations and modelling products (e.g., Siokou-Frangou et al., 2010; Varkitzi et al., 2020; D'Ortenzio et al., 2021; Habib et al., 2023), while the 2022 event was located southeast of Crete (Fig. 1). In this work we analysed and described the 2022 event main features traits and its drivers."

**2. The authors highlight that the location of the bloom is not the Rhodes gyre, where the cyclonic circulation and the convection often trigger phytoplankton accumulation. Indeed in their map on fig. 1 the South-West border of Rhodes gyre displays low biomass. One then wonders which 3D dynamics was active so to produce a localized bloom. If the forcing was the strong negative heat flux, this should have acted over the whole area. Why the bloom occurred only in that limited area and there was no bloom in the Rhodes area. Having the model simulations for the whole basin the authors should discuss this aspect.**

The driving mechanism of the event is represented by negative heat fluxes. A significant drop in air temperature is, in fact, observed in the area starting from 10 March according to a cold air intrusion from the north west, as shown in Fig R4. A similar cold spell has been recorded also in January 2022 (not shown) with a consequent first temperature drop. These cooling events resulted in significant sea surface temperature (SST) anomaly especially in the southern Levantine basin (Fig. R5). It is worth noting that the impact of 2022 cold spells on the North-Central Aegean Sea has been recently studied by Potiris et al. (2024), which show that buoyancy losses during the winter 2021–2022 was comparable to those of 1993–1994, 2002–2003, and 2012, which were all years of dense water formation (DWF) in the Aegean Sea. The findings of Potiris et al. (2024) further supports the fact that the 2022 winter and related marine processes can be considered quite anomalous for the Eastern Mediterranean.

Concerning the spatial shift of the event with respect to the Rhode gyre, it is worth noting that the negative SST anomalies appeared in the south of Crete and persisted in the area till

the end of March (Fig. R5, satellite L4 product). Moreover, relatively cold SSTs are also observed by the L3 satellite product (Fig. R6) although only from 12 March (the region is cloudy between 9 and 11 March). Modelling products show that the strong mixing event that started on 9 March and ended on 25 March (Fig. R7) is possibly driven by the cooling, and that the area with the highest mixed layer depths (larger than 1000 m) well overlaps with the later April 2022 bloom.

The anomalous localization of the 2022 bloom can be further supported by comparing the vertical processes at two locations: (i) inside the area of the event and (ii) in the Rhodes gyre area where late winter bloom typically occur ("+" and "x" marker in the first panel of Fig. R3, respectively). The Hovmöller diagram of temperature inside the event area reveals the gradual outcropping of deep water masses that on 25 March reached the surface from 2000 m (Fig. R8a). At the same time phosphate concentration shows a nearly vertical uniform distribution with persistent high values in the surface layer (>0.15mmol/m3) till the beginning of the event (4 April), when the nutrient started to be consumed (Fig R8b). Starting on 4 April, large chlorophyll concentration in the surface and subsurface layer follows the nutrient injection (Fig. R8c). Finally, a transition to stratified conditions with formation of a deep chlorophyll maximum (DCM) occurs from 10 April. The location outside the 2022 event (right column of Fig. R8) shows much less intense and shorter water column mixing with lower phosphate concentration in surface layers (Fig. R8d and e). In the chlorophyll Hovmöller diagram (Fig. R8f), a transition phase (non-negligible surface concentration with subsurface chlorophyll maximum;Lavigne et al., 2015) toward summer stratified DCM conditions is already in place at the end of March.

We believe that the above materials and figures support the claim that the anomalous event occurred in an area different from the usual Rhode gyre region. However, given the limits of imposed by the State of the Planet journal, we propose to:

- Add a new paragraph in the result section that will resume the motivation for the location of the extreme event: "L 103 - The daily maps of AST, SST, SST anomaly, MLD and heat fluxes during March 2022 provided in the Appendix A (Figs. A1-A5) further detail the spatial extent and temporal sequence of the atmospheric and oceanic processes summarised in Fig. 2. Two close significant drops in AST are, in fact, observed in the area (11-14 March and 19-23 March) according to a cold air intrusion from the northwest (Fig. A1). Together with the January cold spell (Fig. 2), the March cooling events resulted in significant negative SST anomalies especially south of Crete, which persisted in the area till the end of March (Fig. A2) with more steady occurrences in the anomalous-event area. Moreover, relatively cold SSTs are also observed by the L3 satellite product (Fig. A3) although only on 7 March and from 14 March onwards (the region is cloudy between 9 and 13 March). Modelling products show that the strong mixing event that started on 9 March and ended on 25 March (Fig. A4) is possibly driven by the cooling, and that the area with the highest mixed layer depths (larger than 1000 m) well overlaps with the April 2022 anomalous bloom. The strong negative heat fluxes into the sea, which occur at the same dates of the cooling events (Fig. A5), further confirm that the driving mechanism of the event is represented by significant heat losses".
- Compact Fig. 2 and Fig. 3 in a unique novel Fig. 2, and provide the Hovmoeller diagrams (Fig. R8) as Fig. 3 with relevant comments: "L 171 - The anomalous localization of the 2022 bloom can be further supported by comparing the vertical

processes at two locations (Fig. 4): (i) inside the area of the event and (ii) in the Rhodes gyre area where late winter bloom typically occur ("+" and "x" marker in Fig. 1, respectively). The Hovmöller diagram of temperature inside the event area reveals the gradual outcropping of deep water masses that on 25 March reached the surface from 2000 m (Fig. 4a). At the same time phosphate concentration shows a nearly vertical uniform distribution with persistent high values in the surface layer (> 0.15 mmol m-3) till the beginning of the event (4 April), when the nutrient started to be consumed (Fig 4c). Starting on 4 April, large chlorophyll concentrations in the surface and subsurface layer follows the nutrient injection (Fig. 4e). Finally, a transition to stratified conditions with formation of a deep chlorophyll maximum (DCM) occurs from 10 April. The location outside the 2022 event (right column of Fig. 4) shows much less intense and shorter water column mixing with lower phosphate concentration in surface layers (Fig. 4b and d). In the chlorophyll Hovmöller diagram (Fig. 4f), a transition phase (non-negligible surface concentration with subsurface chlorophyll maximum; Lavigne et al., 2015) toward summer stratified DCM conditions is already in place at the end of March".

- Insert figures (Fig. R4-R7) in the Appendix A together with a novel figure containing daily maps of heat fluxes in sea water.

**3. In Fig. 23 the authors show the time course phosphate concentration above the nutricline. Why phosphate? Because is considered the limiting nutrient? One wonders if the nutricline was relative to a specific nutrient or all the nutrient profiles overlapped.**

We thank the Reviewer for the comment. We focused on phosphate because it is considered the limiting nutrient for the Mediterranean Sea (Siokou-Frangou et al., 2010). We will add a comment explaining the reason for focusing on phosphate (L 65) in the manuscript and we will explain that nutricline in Fig. 2 and in the related text is indeed the phosphocline (L 125). We will also change the line 113 specifying that phosphocline is computed as the depth of the maximum vertical gradient as done in Salon et al. (2019). Moreover, we will substitute nutricline with phosphocline through the manuscript.

**4. There is a time mismatch between satellite and model. The authors acknowledge this, if it is not a mistype, on lines 175-176. However I do not understand why this " ...provides an assessment of the capability of the prediction chain to simulate specific events". Do the author mean that the assessment suggests that the model did not simulate the event correctly? If so why they are mostly relying the simulations in discussing the event? A clarification would help.**

We agree with the Reviewer that the statement was misleading. In reality, we meant the opposite. As explained in the reply to comment 1, the Med-MFC model allows us to describe the whole chain of processes of the event: i.e., from the cold spell events to the evolution of the bloom. The occurrence of the simulated bloom (location, timing and intensity) is consistent with the available observations. We use the term *consistent* meaning that observations support the simulated results that the bloom event occurred in an area outside

the typical Rhode gyre area and with nearly one-month delay with respect to what usually occurs in that area (Fig. R1 and R2). Nevertheless, it is also worth acknowledging that the model simulates the event with a possible delay of 5-8 days (even if the cloud coverage prevents a precise definition of the spatial and temporal extent event). This sentence will be rephrased and made clearer:

"L 161 - Since the 2022 anomalous surface bloom is the result of a sequence of processes (cold spell, sea surface cooling, vertical mixing, fertilisation and subsequent stratification), uncertainties in the representation of each of these dynamics by the atmospheric-ocean and biogeochemical models may combine and result in inaccuracies in the spatiotemporal representation of the bloom. However, even if the bloom simulation shows a delay of 5-8 days, the use of three-dimensional modelled data allowed to: (i) define the temporal and spatial boundaries of the event, and (ii) tackle the sequence of physical and biogeochemical processes that are involved in the bloom dynamics."

**5. Assuming that the model simulation captured to a reasonable extent the dynamics of the mixed layer, Fig. 2 shows that during the bloom time there were three, if not four, events of deep remixing of the water column. This questions the simple reconstruction of the bloom as convection-nutrient upward transport-surface stabilization. Could the authors analyze the dynamics in more detail?**

Figure 2 in the submitted manuscript shows the maximum MLD in the area. As it is made evident in Fig. R7 there is a deepening of the MLD in March in a wide area corresponding to where the bloom occurs, while in April 2022 this is not the case. There is a very local MLD maximum around 25°E - 33°N (Fig. R9) that is depicted also in the Fig. 2 time series (submitted manuscript) and that is a bit misleading in this respect. We will use the mean MLD in Fig. 2 (Fig. R10) and modify the text accordingly to clarify that the mean MLD in the area of bloom reaches its maximum at the end of March 2022:

"L 98 - According with the relatively low SST and similarly to the typical winter mixing conditions in the Rhodes gyre area (Kubin et al., 2019), in the 2022 event area (Fig. 1) the mean mixed layer (MLD; calculated as depth where the density increases by 0.01 kg m-3 compared to density at 10 m depth; product ref. 4, Table 1) is deeper than 500 m (Fig. 2b) on several occasions from the end of January to the end of March, when the mean MLD gets shallower (up to 50 m). Consistently with the strong March 2022 sea surface cooling, the mean MLD reaches its maximum in March (equal or deeper than 700 m)."

**6. The authors mention that "..the local fishery community reported increased catches.." but they do not say where and when. The bloom is quite far from the coast. Did the catches increase in the high sea?**

We have been told that as personal communication from a Greek colleague, however since we have not been able to sustain this information by observations or references, we will remove the sentence from the manuscript.

**The reference Josey, S. and Schroeder, K.: https://doi.org/10.5194/egusphere-egu23-5884, 2023 is never cited in the text.**

Thanks for spotting the unused reference. We will remove it.

Finally, we corrected some typos and introduced small corrections to improve the manuscript readability.

**Reviewer #2**

Our point-by-point responses together with propositions for manuscript updates (in green) are provided below. Figures are inserted at the end of the document while line numbers refer to the new version of the manuscript. In addition to what described below, we corrected some typos and introduced small corrections to improve the manuscript readability.

**As the editor, I hereby provide a review of your submitted paper.**

**I do not find evidence in this paper that there actually was a phytoplankton bloom anomaly in the area in 2022; the existence of such an anomaly can only be demonstrated convincingly when based on observations. In fact, it is stated in the text that ocean colour satellite observations of the area appear to indicate that the bloom location and timing did not occur where the model predicted it to be.**

We agree with the reviewer that the manuscript misses to clearly show that the event is anomalous (in terms of intensity and timing) considering the available observations. The comparison of the available observations with the OC climatology (Fig. R1 and R2) reveals higher-than-usual chlorophyll concentrations as well as an intense bloom that is spatially and temporally shifted with respect to the usual patterns observed in the Cretan area.

In particular, Fig. R1 shows that high chlorophyll concentrations are observed on 27 and 29 March and on 1 and 6 April. On these dates, observed chlorophyll concentration is higher than 0.5 mg m$^{-3}$ (up to 3 mg m$^{-3}$ on 29 March). Moreover, high chlorophyll concentrations are located in an area that differs (southwestern shifted) from the usual "Rhodes gyre" bloom regions, which in Fig. R1 is represented by the magenta contour identifying satellite climatology above 0.115 mg m$^{-3}$ (half of the threshold used to define the event area; Fig 1 in the submitted manuscript). Further, it is worth noting that the area with climatological concentration above the threshold is largest at the beginning of March.

Additionally, an analysis of the deviation of OC observations with respect to the 1999-2020 climatology in the area under investigation demonstrates that on 27 March and 29 March, and on 1 and 6 april chlorophyll is 4 standard deviations higher than the mean (Fig. R2).

For sake of clarity and to include further comparison with observations, daily maps of model surface chlorophyll concentration are provided in Fig. R3. The simulated bloom started on 4 April, reached a peak between 8 and 9 April with concentration larger than 0.5 mg/m3 (i.e., similar values to the ones observed in satellite maps), and gradually extinguished from 11 April onward. On the other hand, from the analysis of satellite maps, it can be presumed that the event started somewhen around 27 March, maintained high concentration values on 29 March, 1 and 6 April and possibly ended between 8 and 9 April.

Even if the simulation shows a delay of 5-8 days, the use of modelled data have some clear advantages since the 3D products allowed to: (i) define the temporal and spatial boundaries of the event, (ii) tackle the sequence of physical and biogeochemical processes that are involved in the bloom dynamics. This second aspect will be strengthened by including additional figures as supplementary material, as proposed to Reviewer #1.

We believe that our contribution is a good example of the capability of an operational analysis and forecasting system to predict marine anomalous events and to provide a consistent and coherent picture of the processes involved using multiple information

sources: atmospheric data, marine physical and biogeochemical modelling results, and observations.

Given the limits imposed by the State of the Planet Journal to the number of figures, we propose the following changes in the new version of the manuscript:

- Substitute old Fig. 4 with Fig. R1b to show the high chlorophyll values observed in OC and their temporal and spatial shift with respect to the area typically impacted by the Rhode gyre bloom.
- Include Fig. R2b and Rb in the novel Appendix A.
- Enrich the result section with a paragraph that highlights how the observations provide evidence of an anomalous event in the area as illustrated above: and the explanation of the observed and simulated sequence of the event as supplementary material in the Results section: "L149 - In particular, Fig. 3 shows that high chlorophyll concentrations are observed on 27 and 29 March and on 1 and 6 April, indicating that the event started around 27 March, maintained high concentration values on 29 March, 1 and 6 April and possibly ended between 8 and 9 April. On these dates, observed chlorophyll concentration is higher than 0.5 mg m-3 (up to 3 mg m-3 on 29 March). Moreover, high chlorophyll concentrations are located in an area that differs (southwestern shifted) from the usual "Rhodes gyre" bloom regions, which in Fig. 3 is represented by the magenta contour identifying satellite climatology above 0.115 mg m-3 (i.e., half of the threshold used to define the event area; Fig. 1). Further, we observed that the area with climatological concentration above the threshold is largest at the beginning of March (not shown). Model daily maps of model surface chlorophyll concentration provided in the Appendix (Fig. A6) show that in the simulation the bloom started on 4 April, reached a peak between 8 and 9 April with concentration larger than 0.5 mg m-3 (i.e., similar values to the ones observed in satellite maps), and gradually extinguished from 11 April onwards. An analysis of the deviation of satellite chlorophyll observations with respect to the 1999-2020 climatology (Fig. A7) demonstrates that on 27 March and 29 March, and on 1 and 8 April chlorophyll is at least 3 standard deviations higher than the mean in the event area" .
- Enrich the discussion section on the mechanism driving late-winter/early-spring blooms in the Levantine basin also considering experimental and modelling studies (e.g., Habib et al., 2023; D'Ortenzio et al., 2021): "L 188 - An anomalous deep mixing and bloom event in the south-eastern Mediterranean in the 2022 late-winter early early-spring period was detected by means of the Copernicus Marine MEDMed-MFC products. In this region, intense phytoplankton blooms related to vertical mixing processes and consequent nutrient supply are usually located in the Rhodes gyre area, and have been previously investigated using in-situ and satellite observations and modelling products (e.g., Siokou-Frangou et al., 2010; Varkitzi et al., 2020; D'Ortenzio et al., 2021; Habib et al., 2023), while the 2022 event was located southeast of Crete (Fig. 1). In this work we analysed and described the 2022 event main features traits and its drivers."

**I would recommend that, instead of using NRT ocean colour products, that you look into time series of Chla from all ocean colour satellites merged together (such as the**

**GlobColour satellite dataset). This will improve spatial and temporal coverage. If still plagued by too high cloud cover, use spatial and temporal averaging to analyse if the 2022 condition did (or did not) present a phytoplankton bloom anomaly.**

We agree with the Reviewer about the need to use observations that provide the highest possible coverage and accuracy. In order to address the suggestion raised by the Reviewer, we updated the maps (new Fig. R1) using the reprocessed multi-year Copernicus Marine Service dataset, that uses better estimates of atmospheric variables with respect to NRT. Both the Copernicus Marine NRT product and the Copernicus Marine reprocessed one merge multi-sensor ocean colour datasets similarly to GlobColour. We also compared chlorophyll maps from GlobColour and Copernicus Marine products without finding relevant differences in the Mediterranean region in terms of  spatial coverage. For instance, for the 29 March the two products ( https://hermes.acri.fr/images/data/EURO/merged/day/2022/03/29/L3m_20220329__EURO_ 1_AVW-MODVIR_CHL1_DAY_00.png and https://data.marine.copernicus.eu/-/4ezccnprsg) have very similar spatial coverage and both show large chlorophyll concentration in the investigated area (up to 3 mg m$^{-3}$).

It is also worth to mention that the Copernicus Marine product used in the manuscript has the advantage of an algorithm specifically developed and tuned for the Mediterranean Sea (details are provided in the documentation at https://data.marine.copernicus.eu/product/OCEANCOLOUR_MED_BGC_L3_MY_009_143/d escription). For this and the above mentioned reasons, we will use the reprocessed Copernicus Marine chlorophyll product in the updated version of the manuscript (new Fig. R1 instead of Fig. 4 of the submitted manuscript), and we will specify that the product merges multi-sensor ocean colour datasets to obtain the highest possible spatial coverage (L 147).

**I consider this issue a major problem and therefore recommend major revisions to your manuscript.**

**I  look forward to seeing a revised version of your work.**

**References**

D'Ortenzio, F., Taillandier, V., Claustre, H., Coppola, L., Conan, P., Dumas, F., Durrieu du Madron, X., Fourrier, M., Gogou, A., Karageorgis, A., Lefevre, D., Leymarie, E., Oviedo, A., Pavlidou, A., Poteau, A., Poulain, P. M., Prieur, L., Psarra, S., Puyo-Pay, M., Ribera d'Alcalà, M., Schmechtig, C., Terrats, L., Velaoras, D., Wagener, T., and Wimart-Rousseau, C.: BGC-Argo Floats Observe Nitrate Injection and Spring Phytoplankton Increase in the Surface Layer of Levantine Sea (Eastern Mediterranean), Geophysical Research Letters, 48, e2020GL091649, https://doi.org/10.1029/2020GL091649, 2021.

Habib, J., Ulses, C., Estournel, C., Fakhri, M., Marsaleix, P., Pujo-Pay, M., Fourrier, M., Coppola, L., Mignot, A., Mortier, L., and Conan, P.: Seasonal and interannual variability of the pelagic ecosystem and of the organic carbon budget in the Rhodes Gyre (eastern Mediterranean): influence of winter mixing, Biogeosciences, 20, 3203–3228, https://doi.org/10.5194/bg-20-3203-2023, 2023.

Lavigne, H., D'Ortenzio, F., Ribera D'Alcalà, M., Claustre, H., Sauzède, R., and Gacic, M.: On the vertical distribution of the chlorophyll a concentration in the Mediterranean Sea: a basin-scale and seasonal approach, Biogeosciences, 12, 5021–5039, https://doi.org/10.5194/bg-12-5021-2015, 2015.

Potiris, M., Mamoutos, I. G., Tragou, E., Zervakis, V., Kassis, D., and Ballas, D.: Dense Water Formation in the North–Central Aegean Sea during Winter 2021–2022, Journal of Marine Science and Engineering, 12, 221, https://doi.org/10.3390/jmse12020221, 2024.

Salon, S., Cossarini, G., Bolzon, G., Feudale, L., Lazzari, P., Teruzzi, A., Solidoro, C., and Crise, A.: Novel metrics based on Biogeochemical Argo data to improve the model uncertainty evaluation of the CMEMS Mediterranean marine ecosystem forecasts, Ocean Science, 15, 997–1022, https://doi.org/10.5194/os-15-997-2019, 2019.

Siokou-Frangou, I., Christaki, U., Mazzocchi, M. G., Montresor, M., Ribera d'Alcalá, M., Vaqué, D., and Zingone, A.: Plankton in the open Mediterranean Sea: a review, Biogeosciences, 7, 1543–1586, https://doi.org/10.5194/bg-7-1543-2010, 2010.

**Figures**

[Figure]

Fig. R1. Daily maps of satellite surface chlorophyll concentration [mg m$^{-3}$] from 1 March (a) to 18 April 2022 (b) with orange line contouring the event area as identified by the analysis and forecast model (Fig. 1 in the submitted manuscript) and magenta line contouring the usual winter blooms in the Rhode area (i.e., satellite chlorophyll climatology equal to 0.115 mg m$^{-3}$, half of the threshold used to delimit the event area; satellite data are from Copernicus Marine Service multi-year product).

[Figure]

Fig. R2. Daily maps of the number of climatology standard deviations over the climatology mean for satellite surface chlorophyll concentration [mg m$^{-3}$] from 1 March (a) to 17 April 2022 (b) with orange line contour of the event area (Fig. 1 in the submitted manuscript). Satellite climatology is from Copernicus Marine Service multi-year product. Light brown areas are without satellite observations.

[Figure]

Fig. R3. Daily maps of model surface chlorophyll concentration [mg m$^{-3}$] from 25 March to 17 April 2022 with orange line contouring the event area (Fig. 1 in the submitted manuscript) and reference points (top left panel) inside ("+" marker) and outside ("x" marker) the event area.

[Figure]

Fig. R4. Daily maps of air temperature from ECMWF IFC atmospheric fields [°C], from 1 to 31 March.

[Figure]

Fig. R5. Daily maps of sea surface temperature anomaly [°C] from 1 to 24 March (L4 Copernicus Marine Service product).

[Figure]

Fig. R6. Daily maps of sea surface temperature [°C] from 1 to 24 March (L3 Copernicus Marine Service product).

[Figure]

Fig. R7. Daily maps of mixed layer depth [m] in March 2022. The area of bloom (green line; Fig. 1 in the submitted manuscript) corresponds to where the water column is mixed down to 2000 m.

[Figure]

Fig. R8. Hovmöller diagrams of temperature in March (top panels), nutrient and chlorophyll concentrations in March and April (middle and bottom panels) inside the event area at 27 °E - 34.02 °N (left panels, white "+" marker in Fig. R3) and outside the event area at 28.5 °E - 35.1 °N (right panels; white "x" marker in Fig. R3).

[Figure]

Fig. R9. Daily maps of mixed layer depth [m] in March-April 2022. The area of the event is highlighted by the green line (Fig. 1 in the submitted manuscript).

[Figure]

Fig. R10. Daily time series - spatially averaged over the event area (Fig. 1 in the submitted manuscript) - from January to May 2022 of (a) air surface temperature (AST), sea surface temperature for satellite (SST satellite, product ref. 6, Table 1) and model (SST Med MFC, 120 product ref. 4, Table 1), and SST satellite climatology; and (b) mean mixed layer depth (MLD: product ref. 4. Table 1) and phosphocline (product ref. 1, Table 1) with climatological percentiles (thin vertical line: 1st and 99th percentiles, thick vertical line: 25th and 75th percentiles, white marker: median).